# Causal-R: A Causal-Reasoning Geometry Problem Solver for Optimized Solution Exploration

**Wenjun Wu**[1,3]**, Lingling Zhang**[1,3]*__**, Bo Zhao**[1,3]**, Muye Huang**[1,4]**, Qianying Wang**[2]**, Jun Liu**[1,3]

[1]School of Computer Science and Technology, Xi'an Jiaotong University, Xi'an, 710049, China
[2]Lenovo Research
[3]Ministry of Education Key Laboratory of Intelligent Networks and Network Security,
Xi'an Jiaotong University, Xi'an, 710049, China
[4]Shaanxi Province Key Laboratory of Big Data Knowledge Engineering,
Xi'an Jiaotong University, Xi'an, 710049, China

## Abstract

The task of geometry problem solving has been a long-standing focus in the automated mathematics community and is drawing growing attention due to its complexity for both symbolic and neural models. Although prior studies have explored various effective approaches for enhancing problem solving performances, two fundamental challenges remain unaddressed, which are essential to the application in practical scenarios. First, the multi-step reasoning gap between the initial geometric conditions and ultimate problem goal leads to a large search space for solution exploration. Second, obtaining multiple interpretable and shorter solutions remains an open problem. In this work, we introduce the Causal-Reasoning Geometry Problem Solver to overcome these challenges. Specifically, the Causal Graph Reasoning theory is proposed to perform symbolic reasoning before problem solving. Several causal graphs are constructed according to predefined rule base, where each graph is composed of primitive nodes, causal edges and prerequisite edges. By applying causal graph deduction from initial conditions, the reachability status of nodes is iteratively conveyed by causal edges until reaching the target nodes, representing feasible causal deduction paths. In this way, the search space of solutions is compressed from the beginning, the end and intermediate reasoning paths, while ensuring the interpretability and variety of solutions. To achieve this, we further propose Forward Matrix Deduction which transforms the causal graphs into matrices and vectors, and applies matrix operations to update the status value of reachable nodes in iterations. Finally, multiple solutions can be generated by tracing back from the target nodes after validation. Experiments demonstrate the effectiveness of our method to obtain multiple shorter and interpretable solutions.

## 1 Introduction

Geometry Problem Solving (GPS) is a long-standing task [30, 10, 27] in the automated mathematics problem solving community [11, 5, 28]. It aims to obtain the final numerical answer with a given geometric diagram and problem text which includes a problem goal (*e.g.*, "Find length of AD."), as illustrated in an example in fig. 1 (a). In recent years, this task has attracted increasing attention due to its challenging requirements of multiple capabilities of the model, including multi-modal understanding, abstract geometric reasoning and mathematical computation.

Existing methods can be categorized into two main branches according to their reasoning mechanisms. Neural-based methods [2, 3, 35] extract the multi-modal features of original geometry problem and

---

*Corresponding author. Email: `nickjunwork@163.com`, `zhanglling@xjtu.edu.cn`

39th Conference on Neural Information Processing Systems (NeurIPS 2025).

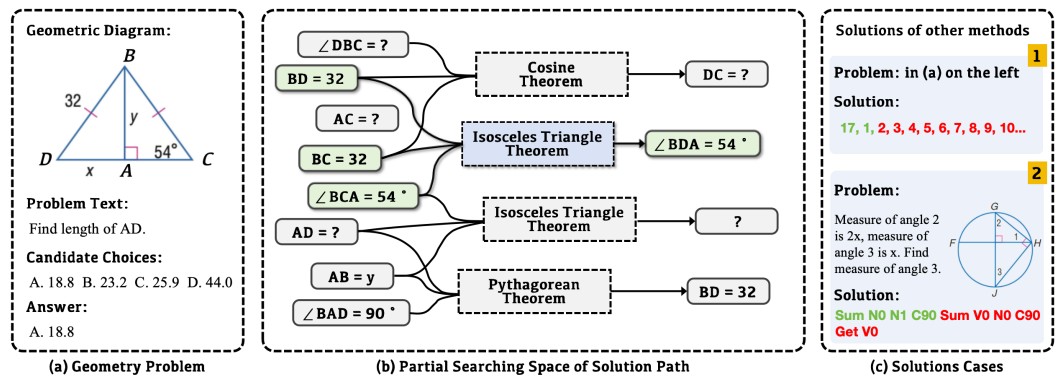

Figure 1: Illustration of solving a typical geometry problem. The red part in solutions in (c) refers to redundant applied theorem indices in case 1 (19) and redundant program sequences in case 2 (35).

predict executable program sequences by neural models, which are then executed by external symbolic engine for the answer. Recent approaches 21, 4 applying large language models primarily distinguish in the output content 9. Symbolic-based methods transform the raw problem into formal language representation 19, 13, 15 which is acceptable by symbolic systems, and continuously apply predefined theorem rules 19, 24, 25 to deduce new geometric conditions until the problem goal is solved. Both approaches have been making efforts on the integration of multi-modal information 18, 20, theorem prediction 39, 14 and development of symbolic systems 23, resulting in significant improvements on problem solving capabilities of the model. However, the application of these models in real-world scenarios 31, such as educational use, is still impeded. We identify the following two overlooked challenges from a broader perspective, which need further exploration from the research community.

First, the multi-step reasoning gap between the initial geometric conditions and ultimate problem goal causes a great search space for solution exploration. Different from general visual reasoning tasks 16, 1, geometry problem solving requires multi-step logical and mathematical deduction, involving many geometric elements and intermediate values that are unknown in the problem. As a result, the search space of potential solution paths becomes extremely large when numerous combinations of deduction paths and geometric conditions are considered (as illustrated in fig. 1 (b)). For complex problems, the search space greatly challenges the pattern-fitting prediction mechanism of neural networks which are trained with only limited geometry problems requiring simple solutions. For symbolic-based approaches, some studies 19, 23 employ pretrained neural models for step-wise theorem prediction to prune the search space in each step, but with limited gains and unreliable performance. E-GPS 31 novelly proposes to solve the problem from the target in a top-down manner, which effectively narrows the search space to a scope which finally leads to the problem goal being solved. Nevertheless, it ignores the constraints of "bottom" known conditions and still searches for paths that should be abandoned from the beginning.

Second, obtaining multiple interpretable and shorter solutions remains an open problem. On one hand, the correctness and interpretability of solutions are not well guaranteed in previous works, impeding the application of models in situations that require solution accuracy (*e.g.*, educational use). Despite leveraging the prediction power of neural models, most prior methods are inherently unable to prevent the introduction of redundant and incorrect steps, even if the final answer is accurate. We list some representative results given by these models in fig. 1 (c). This issue becomes more serious for intricate geometry problems that require longer solution steps, not only increasing the difficulty of problem solving but also decreasing the reliability. On the other hand, obtaining shorter and diverse solutions remains unexplored. For majority of geometry problems, the solution path to problem goal is not unique. Obtaining shorter solution paths, as well as multiple solutions, is beneficial to providing optimized solutions to a problem from various perspectives, enabling the deployment of models in practical scenarios. Simultaneously, it could also address the limitations of existing manually annotated GPS datasets for neural training, using fixed and sub-optimal solution annotations 18, 31.

To overcome the aforementioned challenges, we take a step further into the common process of geometry problem solving, which can be roughly divided into problem understanding, problem

reasoning and problem solving. Interestingly, most works utilize neural networks for the task of reasoning (*e.g.*, theorem prediction in symbolic-based methods or sequence prediction in neural-based methods). Besides E-GPS 31, there is a notable absence of research grounded in symbolic reasoning approaches, where we have identified much potential. Take *Pythagorean Theorem* as an example, if the lengths of two sides of the triangle are known and that one of its interior angles is a right angle, the system could then calculate the length of the third side. Without performing detailed calculation, the process can be simplified as $known \wedge cond \rightarrow result$, where $known$ represents known attributes of primitives (*e.g.*, lengths of two sides), $cond$ represents the prerequisite conditions (*e.g.*, right angle) and $result$ refers to the conditions that are acquirable (*e.g.*, length of the third side). Such causal deduction reasoning process is challenging for neural models to learn, particularly under complicated theorem base and scarce GPS datasets, but is promising for symbolic systems. In this way, the model is able to perform symbolic reasoning by continuously applying the simplified and unified deduction process from the initial geometric conditions, excluding the complicated calculation. Furthermore, the deduction path explicitly records the intermediate outcomes between the start and the end, providing a traceable and explainable reasoning paths.

To achieve this, we propose the Causal-Reasoning Geometry Problem Solver (Causal-R). Specifically, we first follow the common strategy to parse the raw geometry problem into structural formal language representations. Then, we introduce the theory of Causal Graph Reasoning (CGR), where the causal relations between geometric primitives are built in causal graphs according to the predefined theorem base. Each causal graph is a hypergraph consisting of primitive nodes, causal edges and prerequisite edge. Primitive nodes represent the attributes of unique geometric primitives in the parsed conditions, with initial status of *positive* (*i.e.*, value is known) and *negative* (*i.e.*, value is unknown). Causal edge builds the causal relations between head nodes and tail nodes, representing a feasible deduction path. Prerequisite edge points from the prerequisite conditions to a causal edge, controlling its application. In this way, the model is able to perform symbolic-reasoning before actually solving the problem by continuously applying the causal graph deduction until the problem goal is reached. It compresses the search space of solution paths not only from the established conditions at the beginning, but also from the problem goal at the end and the intermediate reasoning paths. Based on the CGR theory, we further introduce the Forward Matrix Deduction (FMD) method which transforms the exhaustive application of causal deductions into matrix operation, enabling the model to perform faster reasoning within iterations. In each iteration, the value vector, representing the reachability of primitive nodes, is updated by matrix operations. When all target values are obtained, the solutions w.r.t. each target can be generated easily by tracking back from the target values to the initial conditions according to the recorded deduction path. These candidate solutions are then sorted in a length-increasing order to verify the feasibility, after which we can obtain multiple solutions to solve the geometry problem. Our contributions can be summarized into the following three points:

- We propose the Causal Graph Reasoning theory which simplifies the theorem deduction as unified causal deduction path for symbolic reasoning. It compresses the search space from the beginning, the end and intermediate process, and ensures the obtaining of optimized solutions.

- We propose the Forward Matrix Deduction method to transform causal graph deduction into matrix operation, enabling the model to perform faster reasoning for solution exploration within iterations.

- We propose the Causal-Reasoning Geometry Problem Solver to verify the above methods. Experimental results demonstrate the effectiveness of our method to obtain optimized solutions, *i.e.*, interpretable, shorter and multiple solutions, in geometry problem solving. Reference of codes is available at https://github.com/nicktech-git/Causal-R.

## 2   Related works

Automated mathematical problem solving has always been a hot topic in the community 11, 5, 28, where geometry problem solving has obtained increasing attention in recent years due to its complexity, requiring multiple crucial capabilities of machine. Earlier methods 7, 8, 32 such as Wu's Method 30 built fundamental approaches for understanding and proving geometry problems. Recent advancements in neural networks and symbolic systems have spurred increasing researches for breakthroughs in problem solving performances of automated geometry problem solvers, which can be categorized into two mainlines.

**Neural-based methods** predict executable or human-readable sequences based on the extracted multi-modal features of raw problem. For example, Chen et al. [2] made the first attempt and established a baseline to uniformly process textual and visual data by encoder-decoder framework and generate program sequences, which are then executed by external symbolic engine. Since then, continuous improvements have been made on the problem types 3, information extraction 35, 18, and cross-modal information alignment 20, 18. With the exploitation of large language models (LLMs), many studies have evaluated and developed the performances of LLMs on mathematic tasks 17, 26, including geometry problem solving 33. Some works focus on improving the multi-modal problem understanding capabilities of multi-modal LLMs 21, especially on geometric diagrams 4, 37, 36. Other works try to enhance the problem reasoning and solving skills of MLLMs by integration of external symbolic engines 22, 6, such as ToRA 12 and MathCoder 29.

However, these methods fall short of guaranteeing the accuracy and interpretability of generated solution paths due to the inherent shortcomings of autoregressive models, likely introducing redundant and wrong steps.

**Symbolic-based methods** continuously apply predefined theorem rules on parsed geometric conditions until the problem goal is solved. Inter-GPS 19 proposes to use a pretrained neural model to predict a sequence of potential needed theorems, improving the efficiency based on brute-search strategy. Subsequently, several works aim at enhancing the capabilities of neural models to analyze the pattern and make improved predictions. For example, Peng et al. [23] and Zou et al. [39] utilize reinforcement learning to enable the model with step-wise prediction power before each step of theorem application. Following them, Huang et al. [14] builds global hologram of geometric conditions and pattern holograms of theorems and uses neural models to predict appropriate theorem based on pattern matching at each step. Recent work Pi-GPS 38 mainly focuses on resolving the textual ambiguities with diagrammatic information, significantly improving the applicability of symbolic-based methods. As an outstanding work, AlphaGeometry 28 highlights the potentials of symbolic models on mathematical proof problems even at Olympic-level, encouraging researchers to keep exploring. E-GPS 31 is the first work that emphasizes the importance of explainability of solutions, advancing the practical application of geometry problem solvers.

However, most of these works mainly focus on solving the geometry problems instead of reasoning for optimized solutions, ignoring the importance of reasonable solutions in real application scenarios. Besides E-GPS, they are unable to ensure the interpretability and accuracy of solutions (*e.g.*, without introducing redundant steps), leaving the exploration of shorter and multiple solutions an ongoing research problem.

## 3 Methodology

Given a geometry problem $(D, T)$, where $D$ and $T$ represent the geometric diagram and problem text respectively, our model aims at obtaining the answer of problem goal $t$ as well as corresponding solutions $S$. The overall framework of our Causal-R is illustrated in fig. 2.

### 3.1 Problem understanding

Understanding the detailed content of multi-modal geometry problem is the basis of following stages. Unlike ordinary multi-modal question answering tasks, solving geometry problems critically relies on the symbolic abstraction and precise mathematical expression. We follow the previous symbolic-based methods 19, 31 to first parse the geometry problem into formalized geometric conditions (*e.g.*, `Equals(LengthOf(BD),32)`), represented in structural clauses, with a commonly used diagram parser 34 and text parser 19. In this way, the problem is converted into a geometric condition set, describing the basic definitions of geometric primitives (*e.g.*, line and angle), non-geometric primitives (*e.g.*, text and symbol), math expressions and their relations.

### 3.2 Causal graph reasoning

#### 3.2.1 Causal graph construction

In this work, causal graph is used to represent the deduction path from known conditions to new acquirable conditions if the prerequisites are met w.r.t. a specific theorem rule. Given the theorem rule base $\mathcal{KB}$ of $K$ rules, $K$ corresponding causal graphs $\mathcal{G} = \{G_1, G_2, \cdots, G_K\}$ are constructed

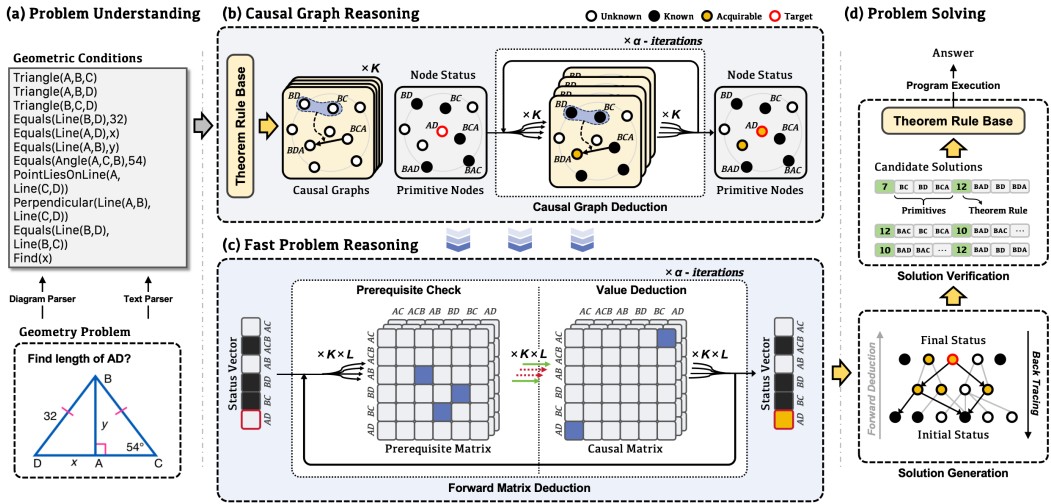

Figure 2: The overall framework of our Causal-R. It mainly contains four parts: (a) in problem understanding stage, the raw geometry problem is parsed into formalized geometric conditions in section 3.1; (b) causal graph reasoning in section 3.2 provides the fundamental theory for causal deduction based on causal graphs; (c) in fast problem reasoning stage in section 3.3, the causal graph deduction is implemented by forward matrix deduction to reason for solutions to final problem goal in iterations; (d) the solutions are generated by tracing back from the target node, and are then verified to obtain the final answer in section 3.4.

according to the detailed theorem knowledge of rules. Each causal graph $G_k$ is a hypergraph consisting of primitive nodes $N = \{n_1, n_2, \cdots, n_M\}$, causal edges $E_k^c = \{e_1^c, e_2^c, \cdots, e_{N_k}^c\}$ and prerequisite edges $E_k^p = \{e_1^p, e_2^p, \cdots, e_{N_k}^p\}$, which are defined as follows:

**Primitive node:** represents the attribute of a unique geometric primitive in the condition set, which stands for visually identified primitive in the geometric diagram. Each node $n_i$ has an initial status (*i.e.*, *positive* and *negative*), where *positive* means the corresponding value of attribute is known or acquirable and *negative* means it is unknown. For example, $\angle$BDA and $\angle$CDB are allocated to the same primitive node with *negative* status. For simplification, we use $\tilde{n}_i$ to represent its status.

**Causal edge:** builds the causal relation between the primitive nodes that points from one or more head nodes to a tail node, denoted as $\tilde{n}_r \leftarrow \mathbf{g}(k, i) \cdot e_i^c(\tilde{n}_a, \tilde{n}_b, \tilde{n}_c)$. When the prerequisite is satisfied, *i.e.*, $\mathbf{g}(k, i)$ is *positive*, $n_r$ is considered reachable and $\tilde{n}_r$ is set *positive* if all status of these head nodes are *positive*. Note that $\tilde{n}_r$ will not be changed into *negative* once it has been changed to *positive*. Here we use $\tilde{n}_a, \tilde{n}_b, \tilde{n}_c$ as placeholder for easier reference and understanding. Please refer to appendix B for detailed formulation of each theorem rule.

**Prerequisite edge:** defines the prerequisite constraints to enable a specific causal deduction, denoted as $\mathbf{g}(k, i) \leftarrow e_i^p(\tilde{n}_a, \mathbf{rel}(n_b, n_c))$, where the prerequisite conditions can be either status of primitive nodes or the relation status of nodes (*e.g.*, length of BD equals length of BC). $\mathbf{g}(k, i)$ is *positive* only when all the prerequisite statuses are *positive*, otherwise *negative*. Please refer to Appendix C for detailed formulation of each theorem rule.

In this way, each $G_k = \{N, E_k^c, E_k^p\}$ records all the possible deduction paths between the geometric primitives in a unified and simplified format according to a specific theorem rule without detailed calculation, which provides the basis for the following reasoning stage. To distinguish the problem goal $t$, we parse it into a series of needed primitive nodes $N^t = \{n_1^t, n_2^t, \cdots, n_Z^t\}, n_i^t \in N$, where the problem goal is reached only if all status of $N^t$ are *positive*.

### 3.2.2 Causal graph deduction

Based on the constructed causal graphs, we are able to perform symbolic reasoning from the initial status of primitive nodes $N$ to a final status of target primitive nodes $N^t$. Theoretically, for an $\alpha$-step geometry problem, the model needs $\alpha$ iterations of causal graph deduction to update all status of $N^t$

to *positive*. For an application of each causal graph $G_k$ in an iteration, all prerequisite edges are first automatically applied to generate the controlling gates $\mathbf{g}(k, \cdot)$ of causal edges. Subsequently, all the causal edges are applied based on the status of primitive nodes from last iteration and the controlling gates $\mathbf{g}(k, \cdot)$ to update the related node status (*i.e.*, all the tail nodes in this causal graph). In each iteration, all causal graphs are applied separately to deduce new reachable node based on the previous status of $N$ from last iteration, so the deduction process of causal graph is self-contained without interference. After each iteration, the status of $N$ is updated according to all causal graph deduction results. Therefore, the causal graph deduction process is able to explore all acquirable primitive nodes after $\alpha$ iterations. The corresponding algorithm is given in appendix A.

To be more specific, we provide a causal graph deduction example in fig. 3 to illustrate how the problem goal is reached regarding the given geometry problem. After two iterations of causal graph deduction, primitive node AD is considered acquirable. By tracing back from node AD, two different solution paths of causal deduction (*i.e.*, 1-A→2-C and 1-C→2-B) can be generated, which corresponds to the actual solutions on the left. It can be seen that the causal graph deduction process includes all possible solution paths but narrows the solution space from the beginning, the end and the intermediate process. On one hand, it constrains the solution path to originate from the known conditions and terminate at the ultimate problem goal, with all intermediate steps adhering to the geometry theorem deduction. On the other hand, any solution path that is feasible to acquire the ultimate answer should be contained in such solution space because it should satisfy these three requirements. It imitates the process by which human experts reason for possible solutions based on the given conditions, but in a full exploration manner.

We highlight three advantages of our causal graph reasoning: (1) **Interpretability:** Since deduction paths are recorded specifically in the causal graphs, it is able for us to trace back for specific causal deduction path and head primitive nodes w.r.t. any acquirable tail node in each iteration, ensuring that the reasoning and solving solution is detailed and explainable. (2) **Multiple solutions:** For a target primitive node, multiple solutions can be obtained by different combinations of feasible causal deduction paths and intermediate nodes. (3) **Shorter solution:** Benefiting from the fact that any feasible solution should be included within the space constrained by our method, it is possible to identify the shortest solution path by tracing back from the first iteration that acquires the target node.

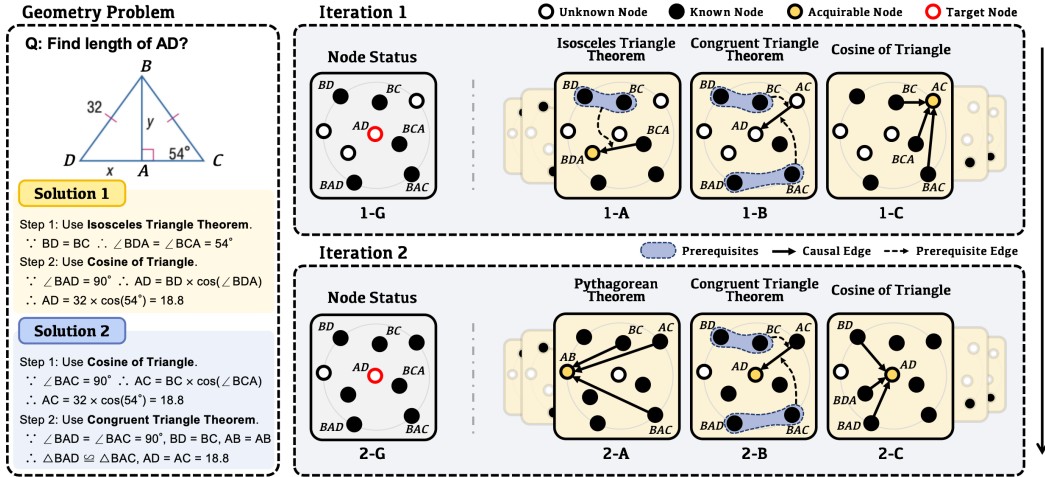

Figure 3: Illustration of 2-iteration causal graph deduction (CGD). Two feasible solution paths of the problem are given on the left, which can be deduced by CGD on the right. For simplification, only the major content is presented in the figure. In iteration 1, the primitive node ∠BDA is acquirable by applying *Isosceles Triangle Theorem* (1-A) on node ∠BCA, because node ∠BCA is *positive* and prerequisite of `rel(BD,BC)` is satisfied in conditions (*i.e.*, BD equals to BC). Similar situation applies to *Cosine of Triangle* (1-C). However, in graph 1-B that represents *Congruent Triangle Theorem*, node AC is still *negative*, causing the failed deduction to obtain node AD. Next in iteration 2, AD is now acquirable due to the *positive* status of either node AC or BDA being just changed in the last iteration.

It can be seen that causal graph reasoning lays the theoretical groundwork for optimized solution path exploration in geometry problem solving. However, the sequential and exhaustive application of causal graph deduction will diminish the advantages of reasoning-solving strategy, making the reasoning process inefficient and cumbersome. To overcome this, we introduce the forward matrix deduction method to realize faster problem reasoning before problem solving stage.

### 3.3 Faster problem reasoning

Moving beyond the aforementioned causal graph reasoning that sequentially applies the causal graph deduction, we novelly propose a faster reasoning method for search space compression. Specifically, the causal graph deduction process is transformed into forward matrix deduction, changing from obtaining the acquirable *positive* status of target node to obtaining its status value.

#### 3.3.1 Matrix definition

To achieve matrix deduction, several matrices are implemented according to the causal graphs $\mathcal{G}$ and primitive nodes $N$. First, an $M$-dimension status vector is designed to represent the status of primitive nodes $N$, denoted as $\mathbf{v} = [v_1, v_2, \cdots, v_M]^{\mathsf{T}}, v_i \in \{-\infty, 1\}$, where $v_i = 1$ means the $\tilde{n}_i$ is *positive*, otherwise *negative*, and $\infty$ is an infinite number. Second, the established relations between the primitive nodes given in the geometry problem are converted into an indicative condition matrix $R^* \in \{0,1\}^{M \times M}$, where $R^*_{ij} = 1$ represents $\mathbf{rel}(n_i, n_j)$ is *positive* otherwise *negative*. Third, the prerequisites of causal graphs are transformed into another unchangeable prerequisite matrix $P \in \{0,1\}^{K \times L \times M \times M}$, where $P_{..ij} = 1$ indicates the relation $\mathbf{rel}(n_i, n_j)$ between primitive node $n_i$ and $n_j$ is required to perform corresponding causal graph deduction, and $L$ refers to number of possible deduction combinations within a causal graph. Fourth, the causal deduction path from head nodes to tail node is recorded by an indicative causal matrix $C \in \{0,1\}^{K \times L \times M \times M}$, where $C_{..ij} = 1$ means the value $v_j$ is needed to obtain the value $v_i$. To specify the target nodes, an indicative target vector $\mathbf{v}^t = [v_1^t, v_2^t, \cdots, v_M^t]^{\mathsf{T}}, v_i^t \in \{0,1\}$ is used, where $v_i^t = 1$ means $n_i \in N^t$.

In this way, all the requirements of causal graph deduction are represented as the positional values in these matrices and vectors, enabling the model to achieve fast reasoning by matrix operation.

#### 3.3.2 Forward matrix deduction

Based on these matrices and status vectors, the causal graph deduction process is transformed into forward matrix deduction for problem reasoning. For an $\alpha$-step geometry problem, the matrix deduction needs $\alpha$ iterations to obtain all values of target primitive nodes. Specifically, in each iteration $o$, an intermediate relation matrix $R^o$ is generated by combining the condition matrix $R^*$ with a value-relation matrix $V^o$ that is calculated by $\mathbf{v}^{o-1}$:

$$R_{i,j}^o = \mathbf{max}(R_{i,j}^*, V_{i,j}^o), V_{i,j}^o = \mathbf{max}(v_i^{o-1}, 0) \times \mathbf{max}(v_j^{o-1}, 0), \tag{1}$$

where $\mathbf{max}(\cdot, \cdot)$ operation keeps the max value of two arguments and $0 \leq i, j < M$. In this way, the relation matrix $R^o$ records whether there is comparable relation, either from the problem conditions or the values obtained from last iteration, between the primitive nodes in iteration $o$. Then, it is used to check whether the existing conditions satisfy the prerequisites by comparing it with the prerequisite matrix $P$ and generate the controlling gate signal $\mathbf{g}(k, l)$:

$$\mathbf{g}(k, l) = \mathbf{max}(1 - \sum_{i=0}^{M-1} \sum_{j=0}^{M-1} \mathbf{max}(P_{k,l,i,j} - R_{i,j}^o, 0), 0), \tag{2}$$

where $0 \leq k < K$ and $0 \leq l < L$. Subsequently, the gate signal $\mathbf{g}(k, l)$ is utilized to control the following process of value deduction, using causal matrix $C$ and previous status vector $\mathbf{v}^{o-1}$:

$$\mathbf{v}_i^o = \gamma + (1 - \gamma) \cdot \mathbf{v}^{o-1}, \gamma = \mathbf{min}(1, \sum_{k=0}^{K-1} \sum_{l=0}^{L-1} \mathbf{g}(k, l) \cdot \mathbf{max}(0, \sum_{j=0}^{M-1} C_{k,l,i,j} \times \mathbf{v}_j^{o-1})), \tag{3}$$

where $\mathbf{min}(\cdot, \cdot)$ returns the minimum of two arguments and $0 \leq i < M$. In this way, the status of acquirable primitive node in this iteration is all recorded in the status vector $\mathbf{v}^o$ and it is feasible to

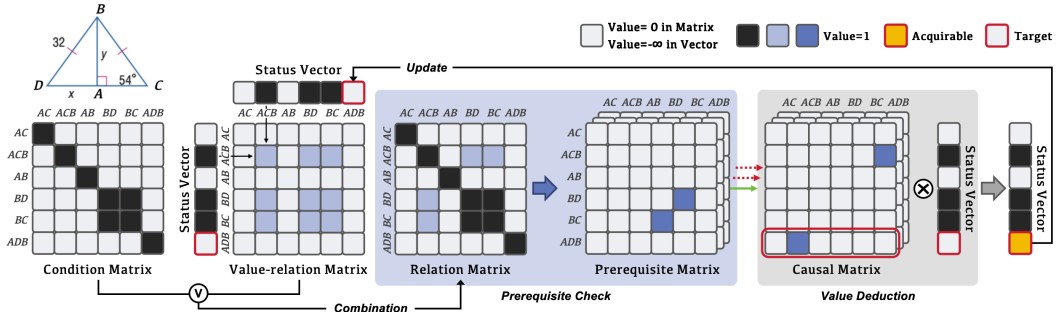

Figure 4: The illustration of forward matrix deduction (FMD) for iteration 1-*Isosceles Triangle Theorem* (1-A) in fig. 3.

check whether the problem goal is reached by verifying the values between $\mathbf{v}^o$ and $\mathbf{v}^t$ using:

$$f(\mathbf{v}^t, \mathbf{v}^o) = \mathbf{max}(0, 1 - \sum_{i=0}^{M-1} \mathbf{v}_i^t \times \mathbf{min}(\mathbf{v}_i^t - \mathbf{v}_i^o, 1)), \qquad (4)$$

where $f(\mathbf{v}^t, \mathbf{v}^o) = 1$ indicates the problem goal is reached, otherwise $f(\mathbf{v}^t, \mathbf{v}^o) = 0$. We give an example of the matrix deduction procedures in fig. 4 to illustrate how the primitive ∠BDA is obtained by *Isosceles Triangle Theorem* in iteration 1 in fig. 3. By applying forward matrix deduction, the update process of primitive node status can be transferred to GPU devices. Benefited from the matrix operation, this method is able to compress the solution search space with an approximate time complexity of $\mathcal{O}(\alpha)$ for a $\alpha$-step geometry problem.

### 3.4 Problem solving

After applying forward matrix deduction, the model completes the process of problem reasoning and obtains a solution space leading to the ultimate problem goal. It is noteworthy that although the forward matrix deduction may acquire many values of those primitive nodes that are not related to problem goal (*e.g.*, new values obtained after $\alpha$ iterations), these values can be easily excluded from the end when tracing back. Theoretically, all feasible solutions are included in this search space, which is recorded in the intermediate results of matrix deduction in iterations. In order to generate shorter solutions, we start from the earliest iteration where each target node is acquired for the first time. Tracing back from the target node with the causal graphs and matrix deduction recordings, a candidate verified solution set $S = \{s_1, s_2, \cdots, s_\lambda\}$ can be obtained, where $\lambda$ is a manually set constraint or the actual number of solutions that exist. Each solution path $s_i$ is an ordered sequence of theorem rule application that is then executed by external symbolic engine for generating the final answer of geometry problem.

## 4 Experiments

### 4.1 Experimental settings

**Dataset.** Following previous symbolic-based methods, we choose the popularly used **Geometry3K** 19 as our benchmark, which includes 3,002 geometry problems that covers a wide range of problem types such as triangles, circles and polygons. Each problem has a geometric diagram and problem text, and is annotated with explicit formal language representations of geometric conditions, providing a friendly platform for symbolic-based methods. For use of neural models, it is split into 2,101, 300 and 601 samples for training, validation and testing respectively.

**Metrics.** For fair comparison, two main evaluation metrics are used: (1) *Accuracy*: A geometry problem is considered correctly solved only if the answer obtained by model is the closest to the ground-truth. (2) *Solution Length*: The average length of solution paths (*i.e.*, the length of theorem application steps) for the solved geometry problems. Each step is vigorously defined in the theorem

Table 1: The main performance comparison results. Solution refers to the *Solution Length* metric. GT means using the annotated parsing results of the problem. Δ means using improved and disambiguated parsing results for reasoning. Methods with a [1] mark and [2] mark use theorem rule base of 17 theorems and 24 theorems, respectively. Human (expert) performances are borrowed from Inter-GPS19.

| Methods | Question Type | | | Geometric Shape | | | | Accuracy | Solution |
| | Measure | Length | Area | Line | Triangle | Quad | Circle | | |
|---|---|---|---|---|---|---|---|---|---|
| Human | 53.7 | 59.3 | 57.7 | 46.7 | 53.8 | 68.7 | 61.7 | 56.9 | – |
| Human Expert | 89.9 | 92.0 | 93.9 | 95.9 | 92.2 | 90.5 | 89.9 | 90.9 | – |
| PGPSNet 35 | – | – | – | – | – | – | – | 77.9 | – |
| LANS 18 | – | – | – | – | – | – | – | 82.3 | – |
| GeoGen-SFT-7B 22 | – | – | – | – | – | – | – | 58.4 | – |
| [1]Inter-GPS 19 | 59.1 | 61.7 | 30.2 | 59.3 | 66.0 | 52.4 | 45.5 | 57.5 | – |
| [1]Inter-GPS 19 (GT) | 83.1 | 77.9 | 62.3 | 86.4 | 83.3 | 77.6 | 61.5 | 78.3 | 7.10 |
| [1]E-GPS 31 | 76.8 | 62.6 | 24.5 | 72.8 | 73.0 | 55.7 | 51.5 | 64.7 | (3.49) 4.19 |
| [1]E-GPS 31 (GT) | 83.8 | 80.0 | 66.7 | 87.7 | 85.3 | 79.2 | 65.9 | 79.8 | (3.35) 3.99 |
| [2]GeoDRL 23 | 75.5 | 70.5 | 22.6 | 77.8 | 76.0 | 62.9 | 53.8 | 68.4 | – |
| [2]GeoDRL 23 (GT) | 86.5 | 93.7 | 75.5 | 87.7 | 93.1 | **90.2** | 78.3 | 89.4 | 2.34 |
| [2]E-GPS 31 | 78.3 | 67.2 | 27.7 | 76.1 | 75.6 | 59.4 | 55.0 | 67.9 | (1.63) 2.28 |
| [2]E-GPS 31 (GT) | 90.4 | 92.2 | 73.6 | 91.4 | 93.1 | 87.9 | 81.1 | 89.8 | (1.57) 2.18 |
| [2]Pi-GPS 38 (Δ) | 83.9 | 81.4 | 59.0 | 79.6 | 83.9 | 76.4 | 73.0 | 77.8 | (2.31) 4.12 |
| [2]Ours | 79.3 | 69.5 | 22.6 | 76.5 | 76.4 | 55.2 | 60.8 | 69.2 | (1.83) 1.98 |
| [2]Ours (GT) | **90.7** | **93.7** | **77.4** | **91.4** | **94.8** | 86.7 | **82.5** | **91.2** | (1.68) **1.89** |

rule base $\mathcal{KB}$, so that the comparison is reasonable. For comprehensiveness, we follow previous works 31, 38 to report the average used theorem rules in the parenthesis under *Solution Length*. However, we contend that this metric lacks significance in identifying shorter solutions, because it merges two steps of the same theorem rule into one.

**Baselines.** For symbolic-based methods, Inter-GPS 19 is the first and representative work that sequentially and continuously applies predefined theorem rules to deduction new geometric conditions. GeoDRL 23 presents a typical collaboration of neural networks and symbolic systems, predicting the theorem to be used each step by neural model trained with reinforcement learning. E-GPS 31 introduces a top-down decomposition mechanism for explainable solution exploration. Pi-GPS 38 adopts LLMs to refine the parsed formal language content for disambiguation and predict appropriate theorems. Additionally, several neural-based methods are selected for in-depth comparison, namely PGPSNet 35, LANS 18 and GeoGen-SFT-7B 22. AlphaGeometry 28 is not considered as our competitor because we handle different kinds of tasks.

**Implementation details.** In consistency with previous symbolic-based methods 31, 23, we use the same theorem rule base provided by Peng et al. [23], which includes 24 predefined rules that represent basic geometry theorems. The maximum constraints of deduction iterations and number of candidate solution paths w.r.t. each target are set to 7 and {1,2,3}, respectively. The early stopping mechanism is used to terminate the iterative deduction process once all target nodes have been reached and the number of iterations exceeds 4. The machine is 24GB NVIDIA GeForce RTX 3090.

## 4.2 Performance comparison

**Quantitative analysis.** Since our method is designed particularly for optimized solution exploration, we mainly focus on the solutions generated by the models, then the problem solving performances. The detailed experimental results are recorded in table 1. We mainly make the following observations: (1) Our method is able to obtain the best *Solution Length* score, *i.e.*, the shortest average solution lengths, compared with all previous methods using the same theorem rule base. Specifically, our Causal-R further shortens the average length of feasible solution steps to 1.89 which is the only one lower than 2. It indicates that Causal-R not only generates more interpretable solutions with less redundancy, but also succeeds in finding shorter solutions at global-level. Although E-GPS also ensures the explainability of solution, it fails to distinguish the shorter solution path when searching from the problem goal. (2) Causal-R achieves the best overall problem solving performances, with

Table 2: The problem solving performances and solution length comparison with different candidate solution constraint. $\lambda$ means the set constraint number of candidate solutions w.r.t. to each target. Best results are in bold.

| $\lambda$ | Accuracy | Solution Length | |
| --- | --- | --- | --- |
| | | Step | Theorem |
| 1 | **91.2** | **1.89** | **1.68** |
| 2 | 90.7 | 1.97 | 1.71 |
| 3 | 91.0 | 1.92 | 1.71 |

Table 3: Comparison of solution quality from five aspects. *Inter.*, *Rea.* and *Solv.* refer to interpretability, reasoning and solving process, respectively. *Mul.* means multiple solutions. ✓ means possessing the quality.

| Methods | Inter. | Rea. | Solv. | Shorter | Mul. |
| --- | --- | --- | --- | --- | --- |
| Inter-GPS 19 | | | | | |
| GeoDRL 23 | | | ✓ | | |
| Pi-GPS 38 | | | ✓ | | |
| E-GPS 31 | ✓ | ✓ | ✓ | | |
| Causal-R (ours) | ✓ | ✓ | ✓ | ✓ | ✓ |

a 1.4% gain on *Accuracy*. Our method is validated as effective across all tested question types and geometric shapes, presenting promising applicability for future extension of symbolic systems. It even surpasses the accuracy of human experts, showing great potentials of symbolic-based methods for GPS. (3) For all symbolic-based methods, the performance is sensitive to the changes of theorem rule base $\mathcal{KB}$ and parsing results of geometric conditions. It not only affects the accuracy of solving geometry problems, but also leads to longer solutions if $\mathcal{KB}$ is smaller. Since developing $\mathcal{KB}$ is easy and a comprehensive $\mathcal{KB}$ is promising for problem solving performance, the key challenge lies in the parsing performance and optimized solution exploration, which is the focus of this work. (4) Neural-based methods are trained with extra GPS data sets and present limited problem solving performances. Furthermore, they usually lack feasible mechanisms to ensure the interpretability of solutions, as well as to find shorter solutions.

To validate the effectiveness of the proposed Forward Matrix Deduction (FMD), we have implemented a basic strategy based on Python dictionaries for Causal Graph Reasoning (CGR). It is noteworthy that CGR is introduced as the theoretical foundation to achieve causal reasoning for solution exploration, and FMD is one of possible detailed strategies that we propose to implement it. Results and according analysis are recorded in appendix C. We conduct further experiments on different number of candidate solution constraint w.r.t. each target node in table 2. Overall, Causal-R presents stable performances when $\lambda$ changes, remaining the best among previous methods in table 1. Surprisingly, the highest accuracy and shortest solution length is achieved when only one solution is needed. We analyze that, on one hand, the symbolic system still exhibits certain randomness such as symbolic assignment order in parsing stage and searching order in back tracing. On the other hand, there is possibility that one target node is the head node of another one. Therefore, acquiring multiple solution has a chance to change the shortest combination of solutions of the two.

**Qualitative analysis.** We further provide a comprehensive comparison between previous methods on the quality of generated solutions from five aspects, as shown in table 3. Both Causal-R and E-GPS are able to ensure the interpretability of solutions, including the explainable reasoning and solving process. However, Causal-R is the only method that is able to generate multiple solutions as short as possible, benefiting from the mechanism of global-level reasoning that acquires the first iteration when target node is achieved. Therefore, our Causal-R possesses the key factors for application in practical scenarios that requires interpretable and accurate solutions. To discuss more profoundly, we have presented several typical cases in appendix D, accordingly with case analysis.

## 5   Conclusion

In this work, we propose a novel model Causal-R to obtain optimized solutions of problem goal from the symbolic-reasoning perspective based on causal graph reasoning theory, which is implemented by the proposed forward matrix deduction method for faster reasoning in iterations. On one hand, Causal-R compresses the search space of solutions from the beginning, the end and intermediate process, alleviating the difficulties for solution exploration. On the other hand, it is capable of obtaining shorter and varied solution paths, while ensuring the interpretability. It presents a promising framework for scenarios where symbolic-reasoning is applicable to. We suggest that our method has potential for generality and applicability to specific problems whose reasoning processes can be approximated into a causal deduction format, beyond the scope of geometry problem solving. However, there are also limitations, which we discussed carefully with feasible solutions in appendix E.

## Acknowledgments and Disclosure of Funding

This work was supported by National Key Research and Development Program of China (2022YFC3303600), National Natural Science Foundation of China (No.62137002, 62293550, 62293553, 62293554, 62437002, 62477036), "LENOVO-XJTU" Intelligent Industry Joint Laboratory Project, the Natural Science Basic Research Program of Shaanxi (2023-JC-YB-593), the Youth AI Talents Fund of China Association of Automation (Grant No._HBRC-JKYZD-2024-311), and the China Scholarship Council Program.

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

## A  Algorithm

---

**Algorithm 1** Causal Graph Deduction

---

**Input:** $\mathcal{G}, N \neq \emptyset, N^t \neq \emptyset, \alpha > 0, o = 1$.
```
/* Initialize the causal graphs, primitive nodes, target nodes maximum
   iteration constraint and current iteration index.                    */
```
**Output:** $N, signal$
```
/* Return the primitive nodes with their updated status and a signal
   indicating whether the problem is solvable.                          */
```
$signal \leftarrow$ False
**while** $o \leq \alpha$ **and** $signal$ **is False do**
    $signal \leftarrow$ True
    $N^o = N$;
    **for** $G_k = \{N, E_k^c, E_k^p\}$ *in* $\mathcal{G}$ **do**
        **for** $e_i^p$ *in* $E_k^p$ **do**
            $\mathbf{g}(k,i) \leftarrow e_i^p(\tilde{n_a}, \mathbf{rel}(n_b, n_c))$ // $\triangleright$Generate the deduction control signal.
        **for** $e_i^c$ *in* $E_k^c$ **do**
            $\tilde{n_e^o} \leftarrow \mathbf{g}(k,i) \cdot e_i^c(\tilde{n_a}, \tilde{n_b}, \tilde{n_c})$ // $\triangleright$Apply the causal deduction.
    $N = N^o$ // $\triangleright$Update the primitive node status.
    **for** $n_i^t$ *in* $N^t$ **do**
        **if** $\tilde{n_i^t}$ *is negative* **then**
            $signal \leftarrow$ False // $\triangleright$The problem target nodes are not fully achieved.
    $o = o + 1$

---

## B  Theorem rule simplification

We provide a detailed illustration of how we simplify the theorem rule into unified causal deduction representation forms in table 4 and table 5. Note that we only list partial transformations w.r.t. each theorem rule for clarity and brevity. For example, in *Angle Sum of Triangle* in table 4, we only list one deduction path, while the deduction path remains valid after the alteration of $n_a$, $n_b$ and $n_c$. Similar situations exist in *Law of Sines* in table 5, too. These deduction paths are extracted from the equations between these values of nodes, where any one of them can be obtained if the other values are known. More detailed implementation of causal graph construction can be found in codes.

Table 4: The simplified causal deduction paths of original theorem rule base. For clarity and brevity, only partial deduction paths w.r.t. each rule are presented. $\Delta$ is used as placeholder to represent the same content within each rule.

| Geometry Theorem/Definition | Simplified Causal Deduction |
|---|---|
| Circle Definition | For radius OA ($n_a$), OB ($n_b$) of circle O:
$\tilde{n_a} \leftarrow e_i^c(\tilde{n_b})$ |
| Thales Theorem | Point A, B, C on circle O, for $\angle$ABC ($n_b$), $\angle$AOC ($n_o$):
$\tilde{n_b} \leftarrow \mathbf{g}(\Delta) \cdot e_i^c(\cdot)$
$\mathbf{g}(\Delta) \leftarrow e_i^p(\tilde{n_o})$ |
| Inscribed Angle Theorem | Point A, B, C on circle O, for $\angle$ABC ($n_b$), $\angle$AOC ($n_o$):
$\tilde{n_b} \leftarrow e_i^c(\tilde{n_o})$ |
| Parallel Lines Theorem | AB // CD, P on AB, Q on CD, for $\angle$APQ ($n_p$) and $\angle$DQP ($n_q$):
$\tilde{n_p} \leftarrow e_i^c(\tilde{n_q})$ |
| Angle Sum of Triangle | For $\angle$ABC ($n_b$), $\angle$ACB ($n_c$), $\angle$BAC ($n_a$) in $\triangle$ABC:
$\tilde{n_a} \leftarrow e_i^c(\tilde{n_b}, \tilde{n_c})$ |
| Isosceles Triangle Theorem (Side) | For AB ($n_{b1}$), AC ($n_{c1}$), $\angle$ABC ($n_{b2}$), $\angle$ACB ($n_{c2}$) in $\triangle$ABC:
$\tilde{n_{b1}} \leftarrow \mathbf{g}(\Delta) \cdot e_i^c(\tilde{n_{c1}})$
$\mathbf{g}(\Delta) \leftarrow e_i^p(\mathbf{rel}(n_{b2}, n_{c2}))$ |
| Isosceles Triangle Theorem (Angle) | For AB ($n_{b1}$), AC ($n_{c1}$), $\angle$ABC ($n_{b2}$), $\angle$ACB ($n_{c2}$) in $\triangle$ABC:
$\tilde{n_{b2}} \leftarrow \mathbf{g}(\Delta) \cdot e_i^c(\tilde{n_{c2}})$
$\mathbf{g}(\Delta) \leftarrow e_i^p(\mathbf{rel}(n_{b1}, n_{c1}))$ |
| Equilateral Triangle Theorem | For AB ($n_{c1}$), AC ($n_{b1}$), BC ($n_{a1}$) $\angle$ABC ($n_{b2}$), $\angle$ACB ($n_{c2}$), $\angle$BAC ($n_{a2}$) in $\triangle$ABC:
$\tilde{n_{a2}} \leftarrow \mathbf{g}(\Delta) \cdot e_i^c(\cdot), \tilde{n_{b2}} \leftarrow \mathbf{g}(\Delta) \cdot e_{i+1}^c(\cdot), \tilde{n_{c2}} \leftarrow \mathbf{g}(\Delta) \cdot e_{i+2}^c(\cdot)$
$\mathbf{g}(\Delta) \leftarrow e_j^p(\mathbf{rel}(n_{a1}, n_{b1}), \mathbf{rel}(n_{a1}, n_{c1}))$ |
| Triangle's Center of Gravity | If Q on AB, M on BC, N on AC, CQ,AM,BN intersects at P, for AQ ($n_{c1}$), BQ ($n_{c2}$), BM ($n_{a1}$), CM ($n_{a2}$), AN ($n_{b1}$), CN ($n_{b2}$), CP ($n_{q1}$), PQ ($n_{q2}$), AP ($n_{m1}$), PM ($n_{m2}$), BP ($n_{n1}$), PN ($n_{n2}$):
$\tilde{n_{q2}} \leftarrow \mathbf{g}(\Delta) \cdot e_i^c(\tilde{n_{q1}}), \tilde{n_{m2}} \leftarrow \mathbf{g}(\Delta) \cdot e_{i+1}^c(\tilde{n_{m1}}),$
$\tilde{n_{n2}} \leftarrow \mathbf{g}(\Delta) \cdot e_{i+2}^c(\tilde{n_{n1}})$
$\mathbf{g}(\Delta) \leftarrow e_j^p(\mathbf{rel}(n_{a1}, n_{a2}), \mathbf{rel}(n_{b1}, n_{b2}), \mathbf{rel}(n_{c1}, n_{c2}))$ |
| Congruent Triangle Theorem (Proving) | For AB ($n_c$), AC ($n_b$), $\angle$BAC ($n_{a1}$), BC ($n_{a2}$) in $\triangle$ABC and DE ($n_f$), DF ($n_e$), $\angle$EDF ($n_{d1}$), EF ($n_{d2}$) in $\triangle$DEF (*SAS*):
$\tilde{n_{d2}} \leftarrow \mathbf{g}(\Delta) \cdot e_i^c(\tilde{n_{a2}})$
$\mathbf{g}(\Delta) \leftarrow e_i^p(\mathbf{rel}(n_c, n_f), \mathbf{rel}(n_b, n_e), \mathbf{rel}(n_{a1}, n_{d1}))$ |
| Congruent Triangle Theorem | For AB ($n_{c1}$), AC ($n_{b1}$), BC ($n_{a1}$), $\angle$ACB ($n_{c2}$), $\angle$ABC ($n_{b2}$), $\angle$BAC ($n_{a2}$) in $\triangle$ABC and DE ($n_{f1}$), DF ($n_{e1}$), EF ($n_{d1}$), $\angle$DFE ($n_{f2}$), $\angle$DEF ($n_{e2}$), $\angle$EDF ($n_{d2}$) in $\triangle$DEF:
$\tilde{n_{f2}} \leftarrow \mathbf{g}(\Delta) \cdot e_i^c(\tilde{n_{c2}}), \tilde{n_{e2}} \leftarrow \mathbf{g}(\Delta) \cdot e_{i+1}^c(\tilde{n_{b2}}),$
$\tilde{n_{d2}} \leftarrow \mathbf{g}(\Delta) \cdot e_{i+2}^c(\tilde{n_{a2}})$
$\mathbf{g}(\Delta) \leftarrow e_j^p(\mathbf{rel}(n_{c1}, n_{f1}), \mathbf{rel}(n_{b1}, n_{e1}), \mathbf{rel}(n_{a1}, n_{d1}))$ |
| Tangent Secant Theorem | AB and circle O tangent at point B, AM intersects circle O at N, for AB ($n_b$), AN ($n_n$) and AM ($n_m$):
$\tilde{n_b} \leftarrow e_i^c(\tilde{n_m}, \tilde{n_n})$ |
| Chord Theorem | A,B,C,D on circle O, AB and CD intersects at point M, for AM ($n_a$), BM ($n_b$), CM ($n_c$), DM ($n_d$):
$\tilde{n_a} \leftarrow e_i^c(\tilde{n_b}, \tilde{n_c}, \tilde{n_d})$ |
| Angle Bisector Theorem | In $\triangle$ABC, M on BC, for AB ($n_{b1}$), BM ($n_{b2}$), AC ($n_{c1}$), CM ($n_{c2}$), $\angle$BAM ($n_{a1}$), $\angle$CAM ($n_{a2}$):
$\tilde{n_{b1}} \leftarrow \mathbf{g}(\Delta) \cdot e_i^c(\tilde{n_{b2}}, \tilde{n_{c1}}, \tilde{n_{c2}})$
$\mathbf{g}(\Delta) \leftarrow e_i^p(\mathbf{rel}(n_{a1}, n_{a2}))$ |

Table 5: The rest simplified causal deduction paths of original theorem rule base, following table 4. For clarity and brevity, only partial deduction paths w.r.t. each rule are presented. $\triangle$ is used as placeholder to represent the same content within each rule.

| Geometry Theorem/Definition | Simplified Causal Deduction |
|---|---|
| Pythagoras Theorem | For AB ($n_c$), AC ($n_b$), BC ($n_a$) in $\triangle$ABC, $\angle$ACB=90°: 
 $\tilde{n_c} \leftarrow e_i^c(\tilde{n_a}, \tilde{n_b})$ |
| Law of Sines | For AB ($n_{c1}$), $\angle$ACB ($n_{c2}$), AC ($n_{b1}$), $\angle$ABC ($n_{b2}$) in $\triangle$ABC: 
 $\tilde{n_{c1}} \leftarrow e_i^c(\tilde{n_{c2}}, \tilde{n_{b1}}, \tilde{n_{b2}})$ |
| Law of Cosines | For AB ($n_{c1}$), AC ($n_b$), BC ($n_a$), $\angle$ACB ($n_{c2}$) in $\triangle$ABC: 
 $\tilde{n_{c1}} \leftarrow e_i^c(\tilde{n_a}, \tilde{n_b}, \tilde{n_{c2}})$ |
| Similar Triangle Theorem (Proving) | For $\angle$ABC ($n_b$), $\angle$ACB ($n_c$), $\angle$BAC ($n_a$) in $\triangle$ABC and $\angle$DEF ($n_e$), $\angle$DFE ($n_f$), $\angle$EDF ($n_d$) in $\triangle$DEF: 
 $\tilde{n_a} \leftarrow \mathbf{g}(\triangle) \cdot e_i^c(\tilde{n_d})$ 
 $\mathbf{g}(\triangle) \leftarrow e_i^p(\mathbf{rel}(n_b, n_e), \mathbf{rel}(n_c, n_f))$ |
| Similar Triangle Theorem | For $\angle$ABC ($n_{b1}$), $\angle$ACB ($n_{c1}$), $\angle$BAC ($n_a$), AB ($n_{c2}$), AC ($n_{b2}$) in $\triangle$ABC and $\angle$DEF ($n_{e1}$), $\angle$DFE ($n_{f1}$), $\angle$EDF ($n_d$), DE ($n_{f2}$), DF ($n_{e2}$) in $\triangle$DEF: 
 $\tilde{n_{c2}} \leftarrow \mathbf{g}(\triangle) \cdot e_i^c(\tilde{n_{b2}})$ 
 $\mathbf{g}(\triangle) \leftarrow$ 
 $e_i^p(\mathbf{rel}(n_{b1}, n_{e1}), \mathbf{rel}(n_{c1}, n_{f1}), \mathbf{rel}(n_a, n_d), \mathbf{rel}(n_{f2}, n_{e2}))$ |
| Similar Polygon Theorem | For AB ($n_a$), BC ($n_b$), EF ($n_e$), FG ($n_f$) in similar polygons ABCD and EFGH: 
 $\tilde{n_a} \leftarrow \mathbf{g}(\triangle) \cdot e_i^c(\tilde{n_e})$ 
 $\mathbf{g}(\triangle) \leftarrow e_i^p(\mathbf{rel}(n_b, n_f))$ |
| Median Line Theorem | Point M on AB and point N on AC, for AM ($n_{m1}$), BM ($n_{m2}$), AN ($n_{n1}$), CN ($n_{n2}$), MN ($n_{a1}$), BC ($n_{a2}$) in $\triangle$ABC: 
 $\tilde{n_{a1}} \leftarrow \mathbf{g}(\triangle) \cdot e_i^c(\tilde{n_{a2}})$ 
 $\mathbf{g}(\triangle) \leftarrow e_i^p(\mathbf{rel}(n_{m1}, n_{m2}), \mathbf{rel}(n_{n1}, n_{n2}))$ |
| Area Equation Theorem | In $\triangle$ABC, point M on BC, AM$\perp$BC, for BC ($n_a$), AM ($n_m$), Area\_$\triangle$ABC ($n_s$): 
 $\tilde{n_s} \leftarrow e_i^c(\tilde{n_a}, \tilde{n_m})$ |
| Angle Sum of Polygon | For $\angle$ABC ($n_b$), $\angle$BCD ($n_c$), $\angle$CDA ($n_d$), $\angle$DAB ($n_a$) in Quadrilateral ABCD: 
 $\tilde{n_a} \leftarrow e_i^c(\tilde{n_b}, \tilde{n_c}, \tilde{n_d})$ |

Table 6: Comparison of time, grouped by number of all deduction paths.

| | $\leq 100$ | $100 - 1000$ | $1000 - 2000$ | $2000 - 5000$ | $5000 - 10000$ | $> 10000$ |
|---|---|---|---|---|---|---|
| **Basic** | $1.57 \times 10^{-3}$ | $1.26 \times 10^{-2}$ | $9.62 \times 10^{-3}$ | $1.31 \times 10^{-2}$ | $2.60 \times 10^{-2}$ | $1.86 \times 10^{-2}$ |
| **FMD** | $1.19 \times 10^{-3}$ | $1.70 \times 10^{-3}$ | $2.35 \times 10^{-3}$ | $6.46 \times 10^{-3}$ | $2.87 \times 10^{-2}$ | $7.76 \times 10^{-2}$ |

## C  Ablation studies of FMD

To provide a fair and detailed comparison, we record the average reasoning time per iteration of these two methods, grouped by the number of deduction paths (i.e., sum of all deduction paths in all causal graphs) in table 6 and the number of unique primitive nodes in table 7. The other parts and settings remain the same. From the results, we mainly have the following observations: (1) FMD consistently performs faster than using Basic strategy when the number of deduction paths is smaller than 5,000 in table 6 and when the number of primitive nodes is smaller than 70 in table 7. Within this range, the time of FMD generally exhibits a steady increasing trend with the increasing of deduction paths and primitive nodes. (2) When the number of deduction paths exceeds 10,000, the time of Basic strategy is surprisingly lower than when the number of paths ranges between 5,000

Table 7: Comparison of time, grouped by number of unique primitive nodes.

| | $\leq 20$ | $20-30$ | $30-50$ | $50-70$ | $>70$ |
|---|---|---|---|---|---|
| **Basic** | $1.23 \times 10^{-3}$ | $4.55 \times 10^{-3}$ | $1.02 \times 10^{-2}$ | $1.93 \times 10^{-2}$ | $4.41 \times 10^{-2}$ |
| **FMD** | $1.19 \times 10^{-3}$ | $1.88 \times 10^{-3}$ | $4.11 \times 10^{-3}$ | $1.65 \times 10^{-2}$ | $6.86 \times 10^{-2}$ |

and 10,000. This is because in this kind of extreme samples, there are only few feasible deduction paths (i.e., successfully activate the path because head nodes and prerequisites are activated) even if there are many connected deduction paths. While for FMD, these paths are still recorded in the matrices and involved in matrix manipulation, resulting in more resource demands. In practice, we simply adopt sequential theorem application of a smaller theorem base that corresponds to feasible causal graphs in each iteration on this problem or the original theorem base, when resources run short. Under such extreme and abnormal circumstances, which is not applicable to most geometry problems, it is recommended to simply use basic strategy of causal graph reasoning.

Moreover, we suggest that FMD can be further optimized by matrix manipulation refinement techniques (we did not adopt any optimization techniques to matrix manipulations in current version). FMD presents great potential for future development in complex scenarios (i.e., scenarios with a higher degree of node involvement), and provides certain insights for similar scenarios in other fields. (Note: A higher degree of node involvement refers to more head nodes and prerequisites required within one causal deduction path. FMD leverages the matrix manipulation and consolidates the sequential condition check into one single operation. This means, regardless of the number of head nodes and prerequisites in a causal deduction path, FMD is able to produce the deducted results at a similar speed. While for the Basic strategy, the needed time increases with the increasing of node involvement degree.)

## D  Case analysis

In order to provide more detailed qualitative analysis, we conduct some cases from Causal-R and present them in fig. 5. In case (a), the model is able to generate two solutions, where the order of theorem application is consistent but the detailed involved primitives are different. Both solutions are reasonable and feasible and lead to correct answer. In case (b), two different solutions with inverted order of theorem applications are obtained. The solutions are simple even if the geometric conditions of the problem seems complex. On one hand, it shows the superiority of Causal-R to explore shorter solutions. On the other hand, it also benefits from the design of symbolic-based strategy which can precisely match the needed conditions within a large set of complicated and redundant geometric conditions. Case (c) is a typical case that our FMD fails to perform matrix manipulation due to resource constraint. The number of deduction paths becomes extremely large when these triangles form different combinations in causal graphs, especially for theorems such as *Congruent Triangle Theorem*. Therefore, the model only performs sequential application of theorems in $\mathcal{KB}$ and obtains the final answer. Case (d) is a failure case that our model fails to obtain the solutions and answer, which mainly attributes to the incorrect parsing of the geometric problem and incomplete design of symbolic system. It remains a difficult task to parse the content with irregular representations such as shaded area. These two cases both indicate the necessity of further developing the symbolic system for adapting to more question types and more theorems.

## E  Limitations

While certain advantages and performance improvements have been obtained by our Causal-R, we also identify some limitations, along with potential solutions, from the following aspects: (1) The effectiveness of causal graph reasoning is influenced by the design of theorem rule base $\mathcal{KB}$. It not only limits the upper bound of problem solving performances (*i.e.*, the geometry problem can not be solved if the needed theorem rule is not contained in $\mathcal{KB}$), but also affects the comprehensive outcome of transformation from theorem rules to causal graphs. This can be resolved by future development and refinement of $\mathcal{KB}$, providing a more standard and larger theorem rule base. Note that once the $\mathcal{KB}$ is determined, the deduction logic for causal graph construction is also determined (such as presented in table 4 and table 5) and does not change for different geometry problems. (2)

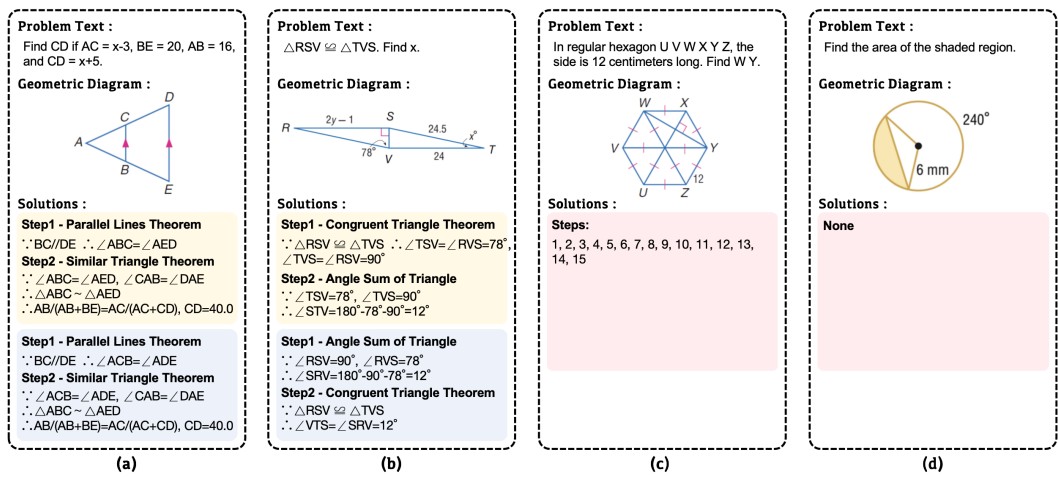

**Figure 5:** Four typical cases from Causal-R. (a), (b) are successful cases and (c), (d) are failure cases.

For more intricate geometry problems, when the combinations of geometric primitives and theorem rules become extremely large, the temporary storage requirement for matrix-based deduction also increases. It can be optimized through both refinement of $\mathcal{KB}$ and application of more efficient matrix operation methodologies. (3) Currently, the method does not support theorem rules that involve constructing geometric primitives (*e.g.*, connect point A and point B as a new line primitive AB). One possible solution is to apply such rules before the causal graph construction to augment the base geometric primitive nodes for a wider solution space that includes the action of constructing new primitives.

