# OpenReview forum: "Causal-R: A Causal-Reasoning Geometry Problem Solver for Optimized Solution Exploration"
_NeurIPS.cc/2025/Conference — NeurIPS 2025 poster_

### Official Review · Reviewer_57ZD · 2025-07-02

**Clarity:** 3
**Significance:** 3
**Originality:** 3
**Rating:** 4
**Confidence:** 3

**Summary:**

This work introduces Causal-R, which solves symbolic geometry reasoning problems with causal graph theory. Based on the causal graph definition, this work also proposes a fast forward matrix deduction algorithm to explore the solution space more efficiently. The authors conducted experiments on the Geometry3K dataset, where the proposed causal graph-based reasoning technique outperforms existing methods in terms of accuracy and solution length.

**Questions:**

See weaknesses.

**Ethical Concerns:**

["NO or VERY MINOR ethics concerns only"]

**Final Justification:**

I would like to maintain my score and a low confidence score as I am not an expert in the field of neural theorem proving.

**Limitations:**

See weaknesses.

**Quality:**

3

**Strengths And Weaknesses:**

**Strength**
- Building a connection between causal graph deduction and geometric reasoning is a promising direction and an important topic.
- The proposed algorithm seems reasonable and new to me. (But I am not an expert in automated theorem proving)
- The paper is well-written and not hard to follow.

**Weaknesses**
- I feel confused about the related works and existing literature, and have the following questions:
1. What is the advantage of using the causal graph deduction against using the problem solver for one-step fact deduction (e.g., LEAN)? Is it that the graph deduction can be formulated as forward matrix deduction (efficiency)?
2. There has been some work [1] discussing converting symbolic rules into graph operations in the symbolic planning context, what is the key difference between this work and [1]? (especially, to me, the forward matrix deduction seems quite like the planning graph expansion process)
3. I agree that AlphaGeometry is not a competitor to this work since you are training-free, so maybe Causal-R is a complementary work to AlphaGeometry (which is based on LEAN). But I would like to see some more explanations about "we handle different kinds of tasks", what kinds of tasks do you and AlphaGeometry consider, respectively?

In my opinion, the related works should be presented in the main paper for a complete understanding of your work and existing works (instead of putting to Appendix).

[1] Mao, Jiayuan, et al. "What planning problems can A relational neural network solve?." Advances in Neural Information Processing Systems 36 (2023): 59522-59542.

---

> ### Author Rebuttal · Authors · 2025-07-31
>
> First of all, we sincerely appreciate your affirmation of our paper and novelty. It encourages us to continuously improve our work. Here are the responses to your questions and concerns.
>
> Q1: What is the advantage of using the causal graph deduction against using the problem solver for one-step fact deduction (e.g., LEAN)? Is it that the graph deduction can be formulated as forward matrix deduction (efficiency)?
>
> First, we want to clarify the differences between our work and works using LEAN such as AlphaGeometry. This is explained in details in our responses to your Q3, so please kindly refer to Q3.
>
> In our work, the core idea of proposing causal graph deduction is that we can abbreviate the high-order geometry theorem rule into unified causal deduction, removing the detailed mathematical calculation in reasoning stage (discussed between L76-87). Each causal deduction path implies a reasoning path that represents a fake theorem application. During this stage, no actual fact deduction is performed and the values in the environment remain unchanged, enabling us to perform global reasoning for solution exploration without actually changing the environment. This process simulates the reasoning stage of neural networks when obtaining the solutions to a geometry problem, but is implemented as symbolic reasoning.
>
> Q2: There has been some work [1] discussing converting symbolic rules into graph operations in the symbolic planning context, what is the key difference between this work and [1]? (especially, to me, the forward matrix deduction seems quite like the planning graph expansion process) [1] Mao, Jiayuan, et al. "What planning problems can A relational neural network solve?." Advances in Neural Information Processing Systems 36 (2023): 59522-59542.
>
> Thank you for providing this work. So far as we know, this work is greatly different from ours from the task, the method and the contribution.
>
> 1. The task of the provided work is to analyze the circuit complexity of a Relational Neural Network (RelNN) when it is used as a policy for planning problems. They try to provide theoretical guidance of the circumstances under which a RelNN-based policy can be successfully trained. Our task is geometry problem solving, and we aim at providing a causal-reasoning geometry problem solver for exploration of explainable, shorter and multiple solutions.
>
> 2. The rules and states mentioned in their work also distinguish from ours. The state in their work is represented in the classic STRIPS form and is tied with the detailed scenarios. While we represent the status of node using only positive and negative, which means acquirable/known and unknown, respectively. All nodes have the same indicative status representation format. Similarly, each causal deduction path is abbreviated from the theorem rules in theorem rule base, and involves only the status of nodes and relations of nodes. We do not consider any sequence of involved nodes in a deduction path.
>
> 3. The graph in their work is used to represent a planning problem as input for RelNN, where the node refers to object and the edge refers to the relation between objects. However, in our work, each causal graph corresponds to a theorem rule, representing the deduction path between the nodes by edge connection. Additionally, the causal graph is used to perform symbolic reasoning.
>
> 4. We are confused that we haven’t found the mentioned planning graph expansion process in this work. However, in our Forward Matrix Deduction (FMD), the matrices are transformed from the geometric conditions and all the causal graphs. The status vector is generated from the geometric conditions, where 1 means known (positive) and -unlimited means unknown (negative). During the FMD process, only matrix operation is performed, without integration of neural networks.
>
> Our differences include but not limited to the above points. It is possible that we have some misunderstandings, and we welcome any corrections.
>
> Q3: I agree that AlphaGeometry is not a competitor to this work since you are training-free, so maybe Causal-R is a complementary work to AlphaGeometry (which is based on LEAN). But I would like to see some more explanations about "we handle different kinds of tasks", what kinds of tasks do you and AlphaGeometry consider, respectively?
>
> Our work mainly distinguishes from theirs from but not limited to the following aspects:
>
> 1. Different Task: AlphaGeometry is an automated geometry problem proving system designed for Olympic-level geometry proving problems, which rarely involve calculation of detailed values of primitives. Their task is to prove certain geometric relation between primitives. Our task is to develop a geometry problem solver for solving geometry problems that mainly involve detailed mathematical calculation.
>
> 2. Different Scenarios: The rules in the symbolic deduction engine they adopt are fundamental and basic, particularly designed for each fine-grained step in Olympic-level proving problems. Our work utilizes the higher-order geometry theorems that are applicable in educational scenarios (e.g., Thales Theorem and Law of Cosines). A comprehensive list can be found in table 1 and table 2 in appendix C. These rules are more suitable for educational use.
>
> 3. Different Goal: AlphaGeometry aims at increasing the problem proving accuracy to demonstrate the unlimited potential power of machines on proving extreme difficult geometry problems. Our goal is to propose a geometry problem solver for optimized solution exploration, providing explainable, shorter and multiple solutions in problem solving.
>
> 4. Different Methodology: AlphaGeometry utilizes a symbolic deduction engine to exhaustively generate new geometric conditions until reaching a closure. Our Causal-R transforms the detailed mathematical deduction into unified causal deduction and performs parallel causal graph reasoning at global level for solution discovery.
>
> Q4: In my opinion, the related works should be presented in the main paper for a complete understanding of your work and existing works (instead of putting to Appendix).
>
> Thanks for your advice. We agree that the related work part is better to be placed in main content. Since we have discussed a lot of related methods in Introduction part, we chose to put it in appendix in the first submission. We will present it in the main paper in final version.
>
> Finally, we sincerely ask if you could kindly consider increasing the score if most of the concerns have been resolved. Thank you.

---

> > ### Comment · Reviewer_57ZD · 2025-08-06
> > **Response to Author Rebuttal**
> >
> > Thanks for the detailed response; they have partly resolved my concerns. Since I am not an expert in the theorem-proving literature, I'd keep my score and low confidence about the evaluation of this work.

---

> > > ### Author Response · Authors · 2025-08-06
> > >
> > > We appreciate you taking the time in reviewing our paper and providing these valuable suggestions. Your recognition and support of our work have been a great encouragement to us. We will ensure that the required modifications of the manuscript are appropriately updated in final version.
> > >
> > > Thank you.

---

### Official Review · Reviewer_SAZz · 2025-07-03

**Clarity:** 3
**Significance:** 2
**Originality:** 3
**Rating:** 4
**Confidence:** 2

**Summary:**

The paper presents a symbolic reasoning framework, Causal-R, designed to address the challenge of generating reasoning solutions for geometry problem solving (also termed as GPS in the paper). The authors mention key limitations in existing neural and symbolic methods, particularly regarding solution redundancy, lack of global reasoning, and limited interpretability. To overcome these challenges, Causal-R makes use of causal graph reasoning theory and introduces a forward matrix deduction method that helps in efficient and iterative deduction by reducing the search space, specifically in the problem goal and intermediate reasoning steps, resulting in faster and more accurate inference. Another feature highlighted by the author includes the ability to generate multiple valid and minimal-length solutions.

The authors validate their approach through a list of experiments, comparing Causal-R against both symbolic and neural baselines. The empirical results highlight the proposed method, Causal-R, achieving the highest accuracy (91.2%) and the shortest average solution length (1.89 steps) among all tested methods, outperforming even human experts in some cases. Furthermore, the model maintains stable performance under different constraints on the number of candidate solutions, indicating its robustness and deterministic reasoning capability. A qualitative evaluation is also provided in the main paper for solution interpretability, reasoning clarity, problem-solving effectiveness, minimal solution length, and the ability to generate multiple valid solutions. These results show the model’s practicality and generalizability, suggesting future works for broader application in symbolic reasoning tasks beyond the geometry domain.

**Questions:**

* It seems the causal graph reasoning is the primary reason for improvements. The paper also emphasizes the importance of causal graph reasoning, but there seem to be very limited technical details on how the causal graph is derived from parsed geometric representations. It would be great if the authors could provide a formal or step-by-step description (or pseudocode) of this construction process. Also, if the authors could provide some lines on the accuracy/robustness of this formulation, it would help clarify the process more.

* The forward matrix deduction is claimed to be one of the key innovations/novelty, however, its specific contribution to performance is not isolated in the experiments (which can be done using a set of ablation experiments). I was wondering if the authors tried any controlled comparison to evaluate performance with and without this component? This would help clarify its standalone value within the overall Causal-R framework.

* Another claim regarding the advantage of Causal-R is its ability to generate multiple minimal-length solutions, but the evaluation lacks a metric that captures solution diversity or optimality across those candidates. Will it be possible to define or introduce a quantitative measure for multi-solution quality (e.g., minimality score, or maybe a diversity index in a controlled set of experiments), and report results under it? I guess this (even with a small set of experiments) would definitely provide a broader scope to the proposed approach.

* Another question is regarding the generalizability beyond geometry tasks, though the authors mention it clearly, there are no concrete illustrations or preliminary results outside the domain (which I understand can go beyond the scope of this work). However, if the authors could describe a specific non-geometry task where Causal-R has the potential to be directly applied/tested (some, e.g., that come to my mind include logical puzzles, procedural planning, etc., however, a direct application would need more effort, I guess)? I guess providing even a small case study or illustrative example of just the formulation would help support the generality claim and make it more applicable for future works on similar lines. Since the work seems to have broad potential, it would be good if they were explicitly stated for future work.

**Ethical Concerns:**

["NO or VERY MINOR ethics concerns only"]

**Final Justification:**

After reading the rebuttal and the subsequent clarifications in the discussion, I believe there are a few concerns that remain unaddressed.

While the authors provide their definition of causality (in the rebuttal phase), the framework still lacks a formal line to mainstream causal inference methods (e.g., d-separation, do-calculus, counterfactuals). This, in general, limits the generalizability and theoretical grounding of their "causal" terminology (and may be misleading for the causal literature community). It’s not a critical flaw for the current scope, but should be addressed in follow-up work or clarified more explicitly in the final version. Also, the lack of automated or scalable evaluation for solution diversity and optimality remains a minor gap. The authors acknowledged this in the rebuttal and plan to refine their claims. Given the exploratory nature of this claim, this concern is addressed.

Overall, the primary concern can be addressed with more rigorous treatment of causality (terminologically and theoretically), and with minor revisions and more careful phrasing of certain claims, this work has the potential to be an important contribution to symbolic reasoning in structured domains.

**Limitations:**

Yes.

**Paper Formatting Concerns:**

One of the major formatting concerns is the style of references used in the paper. The paper refers to the related works using numbers; however, there are no numbers in the references section, making it quite difficult to find the exact citation to which the lines in the paper refer. It is recommended to use a style where the citations are easily findable. To be honest, going through the citations and finding what citation is being talked about in the text took too much time and effort in this version. It would be good if the authors could take care of those in the updated version of the paper.

**Quality:**

2

**Strengths And Weaknesses:**

**Strengths**

* The integration of causal graph reasoning into geometry problem solving seems to be a novel direction that distinguishes this work from prior methods, especially those relying on purely neural or symbolic pipelines (though I am not completely through with this line of work, particularly in geometry problem solving). Also, the design of the forward matrix deduction strategy adds another contribution, suggesting a new path for efficient symbolic computation in structured domains like geometry problem solving.

* The presentation quality and clarity are quite good, I would say. The paper is mostly clear in its motivations and high-level design choices (making it easy to follow for someone not from the domain). The authors also do a good job in discussing the broader implications and challenges of symbolic-based systems for geometry problem solving. Tables and qualitative analyses are helpful for understanding the comparative advantages of Causal-R. Also, the empirical evaluation is thorough, showcasing performance gains in both accuracy and solution optimality over several compared baselines, which is helpful for the reliability of the work.


**Weaknesses**

* One of the major weaknesses comes from the causal graph reasoning formalization, since it is highlighted as a key novelty, the paper is expected to provide a formal or step-by-step technical description of how this graph is constructed from parsed geometric inputs. This not only weakens the reproducibility and clarity of one of the most important contributions but also makes it difficult to think whether the proposed method is generalizable to other domains where causal relationships are shaky and how one should construct them.


* Another weakness comes from the lack of some important empirical experiments; for example, the impact of core components such as forward matrix deduction (also one of the primary contributions of this work) is not isolated in the experiments, i.e., it is difficult to validate looking at the current set of results. An ablation study (or even a basic controlled comparison) would significantly strengthen the empirical claims and clarify which elements are most responsible for the improvements.

* At multiple places, there are claims made in the paper that remain open-ended (even after reading the appendix), for example, the Causal-R claims the ability to generate multiple minimal-length solutions; however, there is no quantitative evaluation or diversity metric included to assess this. Including a metric for solution diversity or optimality would make the claim more compelling/rigorous. Though it is understandable that in general these will be valid, stating it in the discussion section would be good rather than in the main contributions/claims/advantages of the listed method, in general, it is expected to justify all the claims made by the method, which, after reading this paper, remain open-ended. A similar claim regarding the generalization to other domains without any specific direction or experiment should be toned down for better presentation quality of the paper.

---

> ### Author Rebuttal · Authors · 2025-07-31
>
> Q1:  Is there causal graph reasoning formalization or step-by-step technical description of how this graph is constructed from parsed geometric inputs.
>
> Thanks for your advice. We have presented how the causal graph is constructed in section 3.2.1 (Causal graph construction) and formal description of explicit theorem rule simplification in appendix C. Based on these, here we provide a more clear guidelines.
>
> For a geometry problem, the unique primitive nodes are generated from geometric conditions (e.g., Line AB and Angle ABC). Initially, all the node status are negative.
>
> Next, for each causal graph, we match the causal deduction of causal graph with the existing primitive nodes for combinations that satisfy the constraints. In this step, the edges from head nodes to the tail node are specified, and if there are prerequisites, the prerequisite edges are also specified. Initially, all the edges are unactivated (negative status).
>
> Then, we check the geometric conditions to change the initial status of primitive nodes. For those that are provided with detailed values, we change their status to positive. Consequently, the status of relations of nodes are also specified according to the status of primitive nodes and geometric conditions (e.g., Line AB equals to Line BC).
>
> Finally, the target nodes are specified according to the existing primitive nodes and problem goal.
>
> We hope the above information addresses your questions.
>
> Q2: Ablation of Forward Matrix Deduction (FMD) is needed.
>
> Thank you for pointing out this. We initially thought that it is not appropriate to compare the efficiency (e.g., time) of using and not using FMD, because the resource consumption is different. However, we find it still necessary to better support the introduction of FMD.
>
> First, we want to re-emphasize that Causal Graph Reasoning (CGR) is proposed as the theoretical foundation to achieve causal reasoning for solution exploration. FMD is one of possible detailed strategies that we propose to implement CGR. So it’s not simply an A+B module-combining architecture that supports independent ablations. We implemented a basic strategy, leveraging Python dictionaries for causal graph reasoning and node status storage. To provide a fair and detailed comparison, we record the average reasoning time per iteration of these two methods, grouped by the number of unique primitive nodes and the number of deduction paths (i.e., sum of all deduction paths in all causal graphs). The other parts and settings remain the same. Experiments were run for 3 times and the average result is reported.
>
> Grouped by number of all deduction paths (/s):
>
> | | [0, 100]  | (100, 1000]  | (1000, 2000]  | (2000, 5000]  | (5000, 10000] | (10000, )  |
> | :--- | :--- | :--- | :--- | :--- | :--- | :--- |
> | **Basic** | 1.57e-3 | 1.26e-2 | 9.62e-3 | 1.31e-2 | 2.60e-2 | 1.86e-2 |
> | **FMD** | 1.19e-3 | 1.70e-3 | 2.35e-3 | 6.46e-3 | 2.87e-2 | 7.76e-2 |
>
> Grouped by number of unique primitive nodes (/s):
>
> | | ≤20 | 20~30 | 30~50 | 50~70 | ≥70 |
> | :--- | :--- | :--- | :--- | :--- | :--- |
> | **Basic** | 1.23e-3 | 4.55e-3 | 1.02e-2 | 1.93e-2 | 4.41e-2 |
> | **FMD** | 1.19e-3 | 1.88e-3 | 4.11e-3 | 1.65e-2 | 6.86e-2 |
>
> From the results, we mainly have the following observations:
>
> (1) FMD consistently performs faster than using Basic when the number of deduction paths is smaller than 5,000 in table 1 and when the number of primitive nodes is smaller than 70 in table 2. Within this range, the time of FMD generally exhibits a steady increasing trend with the increasing of deduction paths and primitive nodes. When the sum of deduction paths increases to 10,000 and even greater, the dimensions of matrices will expand, resulting in more resource demands. Under such extreme and abnormal circumstances, which is not applicable to most geometry problems, it is recommended to simply use basic strategy of causal graph reasoning.
>
> (2) When the number of deduction paths exceeds 10,000, the time of basic strategy is surprisingly lower than when the number of paths ranges between 5,000 and 10,000. This is because in this kind of extreme samples, there are only few feasible deduction paths (i.e., successfully activate the path because head nodes and prerequisites are activated) even if there are many connected deduction paths.
>
> These experiments and according analysis will be added in final version.
>
> Q3: There are some claims made in the paper that remain open-ended, for example, the Causal-R claims the ability to generate multiple minimal-length solutions.
>
> Thank you for the helpful advice. We want to first clarify that we know “minimal” or “shortest” is a very definite term, so we only claim to obtain “shorter” solutions as presented in our paper. On one hand, obtaining the shortest solution is not critically ensured due to certain randomness of current symbolic system (which we already discuss in L323-330). On the other hand, we do obtain shorter solutions according to experiments in table 1, for both applied theorem length and solution step length. As for multiple solution, our Causal-R is designed to inherently possess the abilities for multiple solution exploration, because the causal reasoning is performed in a global manner that includes all feasible solution paths. Note that, this multiple solution is defined on the basis of explainability, so that the multiple solutions are meaningful. We use the parameter lambda to control the maximum number of feasible solutions obtained w.r.t. each target node; however, returning fewer solutions is also accepted, as not all problems have multiple solutions in practice.
>
> For more comprehensive evidence, we have sampled 30 geometry problems to evaluate the results obtained by our method:
>
> — We manually checked whether there is any shorter step to obtain the correct final answer, using the theorems that are contained in the theorem rule base. All solutions are verified to be the shortest.
>
> — We exhaustively applied the theorem rules by all combinations (i.e., 1+2, 1+3, …, 2+1, 2+3, …, 1+2+3, ….) and obtain the shortest applied theorem length that could lead to correct answer. 2 out of 30 are obtained with shorter length, where the applied theorem lengths are the same. Our solution is longer because we consider applications of the same theorem rule on different primitives different.
>
> — We manually checked that 5 out of 30 problems have 2 solutions obtained by our model, and the solutions are reasonable.
>
> As for a new metric to evaluate the diversity, we are afraid that it is currently not feasible. The challenge arises not only from the difficulty of determining whether a problem actually has multiple solutions or not (ground-truth labeling), but also from the lack of related design in baselines.
>
> Q4: A similar claim regarding the generalization to other domains without any specific direction or experiment should be toned down for better presentation quality of the paper.
> Thanks for your suggestion on this. Our motivation on this point is to inspire more researchers from domains where the task can be reformulated as such, especially for those relying on explicitly pre-defined action space and costly real action application. We have recognized that stating such conclusion might not be appropriate and we will revise the related content to present a more rational discussion in final version.
>
> Q5: Another question is regarding the generalizability beyond geometry tasks, though the authors mention it clearly, there are no concrete illustrations or preliminary results outside the domain (which I understand can go beyond the scope of this work). However, if the authors could describe a specific non-geometry task where Causal-R has the potential to be directly applied/tested? Since the work seems to have broad potential, it would be good if they were explicitly stated for future work.
>
> Thank you for your interest in our method and suggestions. We understand that providing related experiments or illustrative examples (prohibited in rebuttal) could better support the broader potential use of our method. As in our responses to Q4, we decide to refine the claim we made in the original submission into a more rational discussion, so it wouldn’t mislead the readers. Currently, we are not able to give detailed application examples because it needs vigorous technical analysis of feasibility and expertise knowledge of a wide range of research domains, which is really out of the scope. However, here we list some critical principles to evaluate its feasibility in tasks in other domains:
>
> 1. There is a set of independent objects that are specified and determined. These objects should also associate unified indicative status representations (e.g., on and off).
>
> 2. There is a pre-defined action space, where each action is expected to change the status of objects directly based on status of other objects. If there are restrictions of performing this action, the restrictions should be able to be transformed from the status of objects.
>
> 3. The task goal is related to status of certain objects.
>
> We find that using causal reasoning prior to the actual execution of an action is helpful if performing the action is costly. Hope this can assist your understanding.
>
> Q6: Formatting concerns,  the style of references used in the paper. The paper refers to the related works using numbers; however, there are no numbers in the references section. It is recommended to use a style where the citations are easily findable. It would be good if the authors could take care of those in the updated version of the paper.
>
> We are truly sorry for the inconvenience caused to your reading experience. We have also noted that the current reference format is not appropriate, as it uses the default plain style. It will be corrected and aligned with the official requirements in final version.

---

> > ### Author Response · Authors · 2025-08-06
> >
> > Dear reviewer, we appreciate your time and effort in reviewing our paper and providing the detailed comments. We are grateful for these valuable suggestions that will help improve our manuscript.
> >
> > If there's any specific point that needs further clarification and explanations, please kindly let us know. We are committed to addressing your remaining concerns and providing more details.
> >
> > Thank you.

---

> > ### Comment · Reviewer_SAZz · 2025-08-06
> >
> > Thank you for your detailed response. Sorry if it took some time on my side to take a look at the entire rebuttal. After reading the rebuttal in detail, I would like to follow up on a few points:
> >
> > * Q1: On the Causal Graph Reasoning Formalization, thank you for explaining the causal graph construction process in more detail. If I understand correctly, the nodes represent primitive geometric entities, and the edges are formed based on rule prerequisites, deductive logic, and concepts used. However, from a formal causal inference perspective, there's a crucial distinction between syntactic preconditions and semantic causal relationships. Preconditions in a rule base (e.g., "If A and B, then C") can form deductive paths, but these do not necessarily imply causal links unless assumptions of necessity (i.e., the effect does not occur without the cause) and sufficiency (i.e., the cause always produces the effect) are satisfied. I wonder whether this aspect has been considered formally, especially in terms of how your constructed graph would behave if analyzed using causal inference tools (e.g., d-separation, collider/fork/chain structures, or computing average treatment effects, ATE). This raises some interesting questions 1)  How are collider structures (A → C ← B), forks (A ← C → B), and chains (A → B → C) identified and handled within your causal graph structure? 2) Is the system capable of distinguishing between spurious correlations vs. valid causal implications in ambiguous or under-specified problem inputs? 3) Has any form of sensitivity analysis or do-calculus-based reasoning been considered to validate that the edge directions (i.e., causality vs. mere deductive flow) are semantically sound? I believe providing further theoretical grounding or justification for why the graph structure aligns with accepted notions in causal reasoning (e.g., Pearlian or counterfactual frameworks) could significantly strengthen this contribution. Otherwise, some of the terminology might be seen as metaphorical rather than formally causal.
> >
> > * Q2: On the FMD Component and Ablation, thank you for conducting and reporting the comparative efficiency results between the FMD and basic implementation. This is a useful addition. Some of the observed trends (e.g., FMD’s performance degrading when deduction paths exceed 10,000) raise important questions about its scalability, especially if this framework is to be applied in more complex symbolic domains in the future. Including some analysis (or even a brief discussion) of how FMD’s scale and tradeoff would be good to add.
> >
> > * Q3: On the Evaluation of Multiple Minimal-Length Solutions, the discussion on how multiple solutions are explored and how solution lengths are verified was informative. Thank you. Also, manual verification over 30 problems, while useful as a case study, may not provide the generalization power required to substantiate claims regarding minimality and solution diversity. Manual validation can be subjective and is not scalable.  Given the current scope, it might be more accurate to soften the claim from “generating multiple minimal-length solutions” to “capable of generating multiple reasonably short and interpretable solutions,” while leaving space for future work to formalize solution diversity metrics.
> >
> > * Q4 & Q5: On Generalization Beyond Geometry, I am happy that the authors acknowledge the need to rephrase the generalization claims. The response also outlines some key principles (e.g., presence of objects with defined status, pre-defined action space, and causal transitions), which are quite helpful.
> >
> > * Q6: Formatting and Citation Issues. Thank you for acknowledging the citation formatting issue.

---

> > > ### Author Response · Authors · 2025-08-07
> > >
> > > We appreciate your valuable time and effort in reviewing our paper and fostering further discussion. We are delighted to see the interest you have shown in our work.
> > >
> > > **Responses to Q1:** Thank you for the thorough and insightful feedback ! We want to clarify that the ‘causal’ here in our method could be more generalizable and a bit different from the rigorous definition of the formal causal inference. In our work, we emphasize that if the A and B can be obtained (and D can be satisfied if there is prerequisite D), then obtaining C is promising, w.r.t. specific theorem rule. That is to say, A, B and D are all necessary and required for obtaining C. We name it ‘causal’ because in a theorem rule, the detailed value of C is calculated based on the values of A and B, and the acquirable status of C directly depends on the status of A, B and D. They do not possess a real-world causal relationship, but **from the perspective of applying a specific theorem rule**, they form a definite causal result. Therefore:
> > >
> > > (1) The collider structure (A → C ← B) is similar to aforementioned situations, where A and B are both necessary and required for obtaining C. We do not consider the situations where the correlation between A and B exists when C is known in the structure (A → C ← B), because it is specified in other paths (e.g., C → A ← B or C → B ← A). The fork structure (A ← C → B) actually represents two independent causal deduction paths in our work, where C is necessary and required to obtain A, and C is also necessary and required to obtain B. These two deduction paths are not related and can be activated in the same iteration if status of C is positive after the previous iteration. The chain structure (A → B → C) actually indicates two sequential causal deduction paths in our work, where the B should be first obtained by path A → B in some causal graph, and then C can be obtained by path B → C. In this situation, there will be two iterations for the acquiring of C.
> > >
> > > (2) The system does not have the ability to distinguish between spurious correlations vs. valid causal implications, and it does not need to. The generation strategy for causal deduction paths is predefined and deterministic according to specific theorem rules. In cases of ambiguous or under-specified problem inputs, some causal deduction paths may be absent or unable to be activated. The utilization of causal graphs can be seen as the transfer of node status through causal deduction paths.
> > >
> > > (3) As we have discussed before, the causal deduction path is generated according to specific theorem rule. Particularly, the edges are constructed in alignment with the actual calculation formula that is validated to be correct. The edge direction represents the actual flow of node status within such specific theorem rule.
> > >
> > > This question is truly helpful, and has promoted us to carefully add related discussion in final version to better clarify the definition of ‘causal’ in our method.
> > >
> > > **Responses to Q2:** Thanks for the advice. For some extreme cases with more than 10,000 deduction paths, which are uncommon in real life in GPS, most of the paths are actually quiet (i.e., unable to activate). According to our implementations, these paths are simply skipped in basic strategy, while they are not eliminated in FMD, occupying the space and reducing the speed. It can be further modified by matrix manipulation refinement techniques (we did not adopt any optimization techniques to matrix manipulations in current version). Additionally, FMD still holds great potential for future development in complex scenarios (i.e., scenarios with a higher degree of node involvement), and provides certain insights for similar scenarios in other fields. (Note: A higher degree of node involvement refers to more head nodes and prerequisites required within one causal deduction path. FMD leverages the matrix manipulation and consolidates the sequential condition check into one single operation. This means, regardless of the number of head nodes and prerequisites in a causal deduction path, FMD is able to produce the deducted results at a similar speed. While for the basic strategy, the needed time increases with the increasing of degree of node involvement.) We will add according analysis with the comparison results in final version.
> > >
> > > **Responses to Q3:** Thanks for your suggestion. We believe such modification is more accurate, and we will check throughout the manuscript to refine the claims.
> > >
> > > **Responses to Q4-Q6:** We are pleased that our responses are beneficial.
> > >
> > > It would be a great encouragement if you kindly consider increasing the score. Thank you.

---

> > > > ### Author Response · Authors · 2025-08-09
> > > >
> > > > Dear reviewer SAZz,
> > > >
> > > > Thanks for your valuable time and effort in reviewing our paper. With the discussion period drawing to a close, we would appreciate it if you could take a moment to review our responses to your last comments. We hope your concerns have been successfully addressed and would be grateful if you kindly consider updating the score.
> > > >
> > > > Thank you.

---

### Official Review · Reviewer_VP5W · 2025-07-05

**Clarity:** 3
**Significance:** 2
**Originality:** 3
**Rating:** 4
**Confidence:** 2

**Summary:**

This paper proposes Causal-R, a symbolic geometry problem solver based on Causal Graph Reasoning (CGR) and Forward Matrix Deduction (FMD). The method effectively compresses the solution space and generates interpretable, shorter, and multiple solution paths, addressing key limitations of existing approaches.

**Questions:**

1. While Forward Matrix Deduction is proposed to accelerate reasoning, there is little quantitative evidence on runtime or memory usage. How does the reasoning efficiency compare to prior symbolic systems (e.g., Inter-GPS, E-GPS) on complex problems? Please include runtime benchmarks or complexity analysis, especially on problems requiring deeper reasoning chains.

2. How does the current Reasoning LLM / VLM, such as o3, o4, and r1, perform on your task?

3. How does your method perform on mathvista benchmark?

**Ethical Concerns:**

["NO or VERY MINOR ethics concerns only"]

**Final Justification:**

Authors address my main concerns during rebuttal and discussions, including:

1. The ablation of CGR and FMD.

2. The carification of parameters.

3. The scalbility of the KB size.

**Limitations:**

See weaknesses and questions.

**Quality:**

2

**Strengths And Weaknesses:**

Strengths:
1. Reformulates geometric reasoning as causal deduction, which is both elegant and practical.

2. Outperforms baselines in both solution quality and accuracy on Geometry3K.

Weaknesses:
1. A critical limitation is the absence of ablation studies to validate the individual contributions of core components, namely Causal Graph Reasoning (CGR) and Forward Matrix Deduction (FMD). The current experiments only demonstrate the overall performance of the Causal-R framework but fail to isolate the specific impact of CGR on search space compression, FMD on reasoning efficiency, or their synergistic effects. For instance, it remains unproven whether CGR alone outperforms traditional symbolic reasoning in shortening solution paths or if FMD is genuinely responsible for accelerating the deduction process. For example, design ablation experiments comparing variants such as removing CGR and using only traditional symbolic reasoning (e.g., step-by-step rule application as in Inter-GPS), retaining CGR but removing FMD with pure symbolic iterative deduction, and the full method (CGR+FMD); through comparing accuracy, solution length, and reasoning time, the necessity of each module can be clarified.

2. The sensitivity analysis of hyperparameters is insufficient. While the paper mentions setting the maximum number of iterations to 7 and candidate solution constraints (λ) to 1, 2, or 3, it does not investigate how these choices affect outcomes. There is no exploration of whether fewer iterations might miss optimal solutions for complex problems, whether larger λ values (e.g., 4 or 5) could enhance solution diversity without compromising accuracy, or how initializations of the status vector in matrix deduction influence convergence speed. To improve this, test different values of iteration count α (e.g., 3, 5, 7, 10) to observe their impact on solution length and accuracy, verifying the assumption that "the shortest solution is found in early iterations"; expand the range of λ (e.g., 1-5) to analyze the trade-off between solution diversity and redundancy; and test how the initialization of the status vector (e.g., all negative, partial known node initialization) affects convergence speed.

3. Quantification of reasoning efficiency is inadequate. Although FMD is claimed to reduce time complexity to O(α), no concrete runtime data is provided, nor is there a direct comparison with existing methods like E-GPS in terms of inference speed. For practical applications such as real-time educational feedback, measuring efficiency across problem complexities (e.g., multi-step vs. simple deductions) is crucial but absent. To address this, record the average reasoning time of different methods on the same hardware, grouped by problem complexity; and compare the time consumption and memory usage of Causal-R and E-GPS on large-scale problem sets (e.g., 1000 complex problems) to verify the engineering value of FMD.

4. The "optimality" of generated solutions is not rigorously verified. While Causal-R is said to produce "shortest solutions," there is no comparison with human experts’ optimal solutions or the manually annotated "standard solutions" in the Geometry3K dataset. It remains untested whether Causal-R’s solutions are truly shorter than these benchmarks or if they merely avoid redundancy in existing annotations. To fix this, randomly select some problems, invite geometry experts to provide the shortest solutions, and compare their length and correctness with those generated by Causal-R; and compare the length of solutions from Causal-R with the annotated solutions in the dataset to verify the assumption of "surpassing manually annotated suboptimal solutions."

5. Experiments on the scalability of the theorem knowledge base (KB) are missing. The paper notes that expanding KB improves performance but does not test how larger KB sizes (e.g., 30+ theorems) affect search space expansion or solution redundancy. It is unclear whether Causal-R’s causal graph mechanism can effectively suppress rule combination explosions when KB scales, a key claim for its practical applicability. To address this, gradually expand the number of theorems in KB (e.g., 17, 24, 30), test changes in accuracy and solution length, verify the robustness of the method to KB expansion, and analyze whether the "causal graph’s role in inhibiting rule combination explosions" holds.

---

> ### Author Rebuttal · Authors · 2025-07-31
>
> Q1: Ablation of Causal Graph Reasoning (CGR) and Forward Matrix Deduction (FMD).
>
> Thank you for pointing out this. We initially thought that it is not appropriate to compare the efficiency (e.g., time) of using and not using FMD, because the resource consumption is different. However, we find it still necessary to better support the introduction of FMD.
>
> First, we want to re-emphasize that Causal Graph Reasoning (CGR) is proposed as the theoretical foundation to achieve causal reasoning for solution exploration. FMD is one of possible detailed strategies that we propose to implement CGR. So it’s not simply an A+B module-combining architecture that supports independent ablations. We implemented a basic strategy, leveraging Python dictionaries for causal graph reasoning and node status storage. To provide a fair and detailed comparison, we record the average reasoning time per iteration of these two methods, grouped by the number of unique primitive nodes and the number of deduction paths (i.e., sum of all deduction paths in all causal graphs). The other parts and settings remain the same. Experiments were run for 3 times and the average result is reported.
>
> Grouped by number of all deduction paths (/s):
>
> | | [0, 100]  | (100, 1000]  | (1000, 2000]  | (2000, 5000]  | (5000, 10000] | (10000, )  |
> | :--- | :--- | :--- | :--- | :--- | :--- | :--- |
> | **Basic** | 1.57e-3 | 1.26e-2 | 9.62e-3 | 1.31e-2 | 2.60e-2 | 1.86e-2 |
> | **FMD** | 1.19e-3 | 1.70e-3 | 2.35e-3 | 6.46e-3 | 2.87e-2 | 7.76e-2 |
>
> Grouped by number of unique primitive nodes (/s):
>
> | | ≤20 | 20~30 | 30~50 | 50~70 | ≥70 |
> | :--- | :--- | :--- | :--- | :--- | :--- |
> | **Basic** | 1.23e-3 | 4.55e-3 | 1.02e-2 | 1.93e-2 | 4.41e-2 |
> | **FMD** | 1.19e-3 | 1.88e-3 | 4.11e-3 | 1.65e-2 | 6.86e-2 |
>
> From the results, we mainly have the following observations:
>
> (1) FMD consistently performs faster than using Basic when the number of deduction paths is smaller than 5,000 in table 1 and when the number of primitive nodes is smaller than 70 in table 2. Within this range, the time of FMD generally exhibits a steady increasing trend with the increasing of deduction paths and primitive nodes. When the sum of deduction paths increases to 10,000 and even greater, the dimensions of matrices will expand, resulting in more resource demands. Under such extreme and abnormal circumstances, which is not applicable to most geometry problems, it is recommended to simply use basic strategy of causal graph reasoning.
>
> (2) When the number of deduction paths exceeds 10,000, the time of basic strategy is surprisingly lower than when the number of paths ranges between 5,000 and 10,000. This is because in this kind of extreme samples, there are only few feasible deduction paths (i.e., successfully activate the path because head nodes and prerequisites are activated) even if there are many connected deduction paths.
>
> These experiments and according analysis will be added in final version.
>
> Q2: The sensitivity analysis of hyper-parameters alpha and lambda and how these choices affect outcomes?
>
> In our work, the iteration constraint alpha and candidate solution number lambda are both system parameters that do not affect the method. These values can be settled according to detailed application scenarios and demands.
>
> For the iteration constraint alpha, it is only used to set a maximum iteration for reasoning in case the model keeps running without an end. For majority of the geometry problems, 7 steps is enough to obtain the final answer, while the rest are often unsolvable by the design of symbolic system at this stage (e.g., parsing mistakes and lack of theorem rules). It is a definite result that decreasing the iteration constraint will affect the solving performance, and the number of possible solutions.
>
> For different lambda, the model is signaled to try to obtain lambda solutions to obtain the final answer. So, if multiple solutions are needed, the lambda can be set to greater than 1. In situations that a geometry problem does not have multiple solutions, the model will only return one. Since we generate solutions from the first iteration that all target nodes are reached, there is very small possibility of existing shorter solutions at longer iteration, combing our experiments when lambda is set to 2 and 3 in table 2.
>
> Q3: How initializations of the status vector in matrix deduction influence convergence speed (e.g., all negative, partial known node initialization) ?
>
> Sorry that we do not quite understand this requirement. The initialization of the status vector depends on the given information in the problem, where the value of status vector is 1 means the value of corresponding primitive is given (known) in the original problem. Setting the status vector to all negative indicates no value is provided in the problem, which means the problem is unsolvable (e.g., you can’t get the value of side of triangle without any detailed value). Similarly, setting the status vector to all positive means the problem is already solved. In all cases, changing the status vector from its initialization leads to inconsistency to the original problem and directly affects the results.
>
> Q4: The "optimality" of generated solutions is not rigorously verified. While Causal-R is said to produce "shortest solutions," there is no comparison with human experts’ optimal solutions or the manually annotated "standard solutions" in the Geometry3K dataset. It remains untested whether Causal-R’s solutions are truly shorter than these benchmarks or if they merely avoid redundancy in existing annotations.
>
> Thank you for the helpful advice. We want to first clarify that we know “shortest” is a very definite term, so we only claim to obtain “shorter” solutions in our paper. On one hand, obtaining the shortest solution is not critically ensured due to certain randomness of current symbolic system (which we already discuss in L323-330). On the other hand, we do obtain shorter solutions according to experiments in table 1, for both applied theorem length and solution step length. This is compared using the same predicted input and annotated input without any pre-processing. Further, Inter-GPS didn’t provide any annotation of standard solutions. For more comprehensive evidence, we have sampled 30 geometry problems to evaluate the results obtained by our method:
>
> — We manually checked whether there is any shorter step to obtain the correct final answer, using the theorems that are contained in the theorem rule base. All solutions are verified to be the shortest.
>
> — We exhaustively applied the theorem rules by all combinations (i.e., 1+2, 1+3, …, 2+1, 2+3, …, 1+2+3, ….) and obtain the shortest applied theorem length that could lead to correct answer. 2 out of 30 are obtained with shorter length, where the applied theorem lengths are the same. Our solution is longer because we consider applications of the same theorem rule on different primitives different.
>
> Q5: Experiments on the scalability of the theorem knowledge base (KB), such as gradually expand the number of theorems in KB (e.g., 17, 24, 30), test changes in accuracy and solution length.
>
> We are sorry that we are unable to carry out such experiments because its beyond our scope. Increasing the theorem rule base requires expertise knowledge on geometry theorems and explicit definition of theorems that should be orthogonal to the existing ones. Addition of theorem rules that are able to solve the problems that cannot be solved by old theorem rules is promising to increase the accuracy. Addition of higher-order theorem rules that solve problems with one step rather than combination of several theorems is promising to decrease the solution length. We contend that developing theorem rule base is beyond our scope.
>
> Q6: How does the current Reasoning LLM / VLM, such as o3, o4, and r1, perform on your task?
>
> The existing top-tier large language models do perform well on simple geometry problems, but still present a performance gap on more difficult ones compared to symbolic-based methods. Compared with accuracy performance, there are more severe problems of these LLMs to solve geometry problems. On one hand, the solutions generated are often unreliable (e.g., including incorrect calculated intermediate result and inconsistency of contents) even if the final answer is correct. On the other hand, it is not sure whether there is data exposure issue. We use the edited diagram and problem text (e.g., change of value) to test Gemini-2.5 Pro, and it still generates the similar solution content or different solutions but the same answer as the original one. (We have cleared the history and tested in different windows.)
>
> Q7: How does your method perform on mathvista benchmark?
>
> MathVista is not applicable for the evaluation of symbolic-based GPS methods. It is a dataset mainly to test the math capabilities of LLMs/VLMs. It includes many topics such as numerical commonsense, statistical reasoning and scientific reasoning. There is a small portion of geometry problems but do not provide any formal language annotation. Geometry3K is a widely recognized and well-annotated data set that has provided both predicted and ground-truth formal language parsings, covering a wide range of problem types, many of which are also sampled as composition of other datasets. Therefore, it has been considered as a more suitable benchmark for comparison of symbolic-based methods by previous works.
>
> Finally, we want to emphasize that our main goal and contribution is to provide a symbolic-reasoning method using causal graphs for optimized solution exploration (i.e., explainable, shorter and multiple solutions). We sincerely hope if you can kindly consider the limitations of existing development of the geometry problem solving domain, and increase the score if most of the concerns have been resolved.

---

> > ### Author Response · Authors · 2025-08-05
> >
> > Dear reviewer, we appreciate your time and effort in reviewing our paper and providing the detailed comments. We are grateful for these valuable suggestions that will help improve our manuscript.
> >
> > If there's any specific point that needs further clarification and explanations, please kindly let us know. We are committed to addressing your remaining concerns and providing more details.
> >
> > Thank you.

---

> > ### Comment · Reviewer_VP5W · 2025-08-05
> >
> > Thanks for your detailed rebuttal! I have these follow up questions:
> >
> > 1. Is it possible to list the speed of E-GPS in the table (e.g., Grouped by number of all deduction paths and Grouped by number of unique primitive nodes)?
> >
> > 2. What is the size of KB used in your experiments? If it is not possible to increase the size of the KB for Q5 in your rebuttal, then how about reducing the size of the KB to demonstrate scalability?

---

> > > ### Author Response · Authors · 2025-08-07
> > >
> > > Yes, here we provide more detailed analysis of the performances when the size of knowledge base changes. Based on the original version of 24 theorem rules which we use, we block the usage of 4 rules (i.e., *Similar Polygon Theorem*, *Median Line Theorem*, *Area Equation Theorem* and *Line Construction Theorem*). Next, we further block the usage of 3 rules (i.e., *Triangle’s Center of Gravity*, *Angle Bisector Theorem* and *Angle Sum of Polygon*), resulting in a KB of size 17. Further, we continue to block 2 more rules (i.e., *Law of Cosines* and *Similar Triangle Theorem*). Results are recorded in details in the following table.
> > >
> > > | KB Size | Measure | Length | Area | Line | Triangle | Quad | Circle | Accuracy | Solution |
> > > | :--- | :--- | :--- | :--- | :--- | :--- | :--- | :--- | :--- | :--- |
> > > | 24 | 90.7 | 93.7 | 77.4 | 91.4 | 94.8 | 86.7 | 82.5 | 91.2 | (1.68) 1.89 |
> > > | 20 | 91.1 | 89.7 | 67.9 | 90.1 | 94.4 | 81.1 | 84.6 | 88.5 | (1.57) 1.79 |
> > > | 17 | 85.2 | 88.7 | 77.4 | 87.7 | 92.4 | 79.0 | 83.2 | 85.9 | (1.47) 1.70 |
> > > | 15 | 84.4 | 72.8 | 73.6 | 72.8 | 76.4 | 76.9 | 79.7 | 77.0 | (1.24) 1.45 |
> > >
> > > From the results, it can be seen that both the accuracy and length of solution will decrease as the size of knowledge base is reduced. The decrease in accuracy is attributed to the inability to solve geometry problems that require theorem rules reduced from the knowledge base. For example, we can observe an obvious accuracy loss from 92.4% to 76.4% at Triangle when both Law of Cosines and Similar Triangle Theorem are blocked for solving problems. The average solution length decreases accordingly, because those problems requiring longer solution steps are unable to be solved when the needed theorem rules are absent (especially for those high-order theorem rules such as Law of Cosines). Therefore, it is demonstrated that the model performance can be improved by developing the knowledge base. Moreover, our method provides a systematical convenient approach for blocking or adding according theorem rules and causal graph constructions in the future.
> > >
> > > As for the speed of E-GPS, we are sorry that we are unable to record this result because the concepts of causal deduction path and unique primitive node are proposed in our method and this grouping criterion is designed for our method to validate the effectiveness of FMD. However, here we provide further analysis to highlight the differences between our method and E-GPS. In E-GPS, they decompose the parent target into child targets, according to specific theorem rule, and continuously perform the decomposition in a top-down manner until reaching the bottom known conditions. In this process, these targets are specified w.r.t. a theorem rule and being checked with the original conditions. Even though it restricts the model to start from the problem goal, which reduces the reasoning gap between known conditions and final target, the original conditions do not serve as guidance. Therefore, the model still lacks signal to select appropriate theorem rule for decomposition, which impedes the correct and shorter solution exploration. However, in our method, the unique primitive nodes and their status are specified at the beginning. By causal graph deduction, the model is able to reason from the bottom conditions to final target at a global perspective. Both known conditions and final target serve as guidance and restrictions in this process. It also enables us to generate shorter and multiple solutions that lead to final answer.
> > >
> > > If your concerns have been resolved, please kindly consider increasing the score. It would be a great encouragement to us.
> > >
> > > Thank you.

---

> > > > ### Comment · Reviewer_VP5W · 2025-08-08
> > > >
> > > > Thank you for your reply, which addresses my concerns. I will raise my score.
> > > >
> > > > Best regards,
> > > >
> > > > Reviewer VP5W

---

> > > > > ### Author Response · Authors · 2025-08-08
> > > > >
> > > > > We appreciate you taking the time in reviewing our paper and engaging in the discussion. Your recognition and support of our work have been a great encouragement to us. The mentioned modifications of the manuscript will be updated appropriately in final version.
> > > > >
> > > > > Thank you.

---

### Official Review · Reviewer_kUXM · 2025-07-05

**Clarity:** 4
**Significance:** 3
**Originality:** 3
**Rating:** 4
**Confidence:** 5

**Summary:**

The paper introduces Causal-R, a symbolic-based geometry problem solver that proposes a Causal Graph Reasoning (CGR) theory and a Forward Matrix Deduction (FMD) method to efficiently and interpretablely solve geometry problems. The approach systematically encodes theorem-based deduction paths as causal graphs and matrix operations to reduce the search space and extract interpretable solutions. The paper conduct experiments on Geometry3K, showing improvements over symbolic and neural baselines in both solution accuracy and length.

**Questions:**

See the weakness part for the questions. They are:
1. For example, how does it perform on other popular datasets, such as the recent FormalGeo7k or geometry questions in the OpenAI MATH dataset?
2. In the failure case analysis, Is the method robust to these intermediate error?

**Ethical Concerns:**

["NO or VERY MINOR ethics concerns only"]

**Final Justification:**

After considering the author rebuttal, I have updated my assessment. The authors responded constructively to most of the concerns raised. In particular:
- The overclaim of generality has been acknowledged and will be revised.
- The authors conducted an ablation study to compare FMD with a basic implementation.
- The authors have committed to including a complete worked example and improved formatting, which will enhance interpretability.

However, some issues remain unresolved:
- There is still no empirical robustness analysis, and the evaluation remains limited to a single dataset (Geometry3K), with no experiments on other benchmarks or under noisy conditions.
- The discussion of method-level failure modes (e.g., ambiguous or conflicting deduction paths) is still limited.

Overall, I value their thoughtful engagement and planned improvements, and I have reflected that in my updated score. Nonetheless, the remaining issues limit the paper’s general significance at this stage, and I encourage the authors to fully incorporate the promised enhancements in the final version.

**Limitations:**

While the authors include a brief paragraph on limitations in Section 5 and reference Appendix E, the discussion is incomplete and overly superficial. The listed limitations primarily focus on external components such as diagram/text parsing or the lack of certain theorems in the rule base, but the core limitations of the proposed causal reasoning framework itself are not meaningfully examined. For example, is the 24 hand-crafted symbolic rules brittle in other datasets?

Additionally, the authors assert that their method has potential generality beyond geometry problem solving, but no limitations are discussed regarding this extrapolation, which may mislead readers about its broader applicability.

**Paper Formatting Concerns:**

The citation format in the paper looks strange and hard to follow. First, the references are not numbered in the appendix, making it hard to search for the corresponding reference. Second, the reference number can be wrapped in a square bracket [] for better visibility. Currently, it merges into the main text and makes it less readable.

**Quality:**

3

**Strengths And Weaknesses:**

Strengths:
1. I like the paper that provided a clear problem formulation and motivation. The paper identifies two critical issues in prior works: large multi-step reasoning space, and lack of short and interpretable multi-solution paths.
2. The paper introduce a novel framework that combines the causal graph reasoning and faster forward matrix deduction. FMD is an efficient abstraction of symbolic deduction, making the reasoning process computationally tractable.
3. By design, the solution step maps to a known theorem, aiding interpretability and potential educational application.
4. The experiments demonstrate the good performance of the poposed method on the Geometry3K dataset and even outperforms human expert.

Weakness:
1. The proposed method only has been tested on a single dataset, Geometry3K, which was introduced in 2021, four years ago. There’s no exploration of robustness to noisy inputs, or evaluation on other datasets to test generalizability. For example, how does it perform on other popular datasets, such as the recent FormalGeo7k or geometry questions in the OpenAI MATH dataset?
2. There is no ablation to disentangle the contribution of: causal graph reasoning vs. forward matrix deduction (e.g., performance and efficiency); Predefined theorem base vs. solution exploration strategy; Impact of number of iterations or candidate paths on final performance in more depth.
3. The paper over-claims the method is general beyond geometry problem solving in the section 5 (e.g., “causal deduction format” can be applied broadly) but provides no empirical or theoretical support outside of geometry in this paper. This feels speculative and should be toned down unless supported.
4. The case analysis in Appendix D attributes failure primarily to parsing errors or missing theorems in the rule base (e.g., absence of the Geometric Mean Theorem). However, these do not sufficiently probe fundamental limitations of the proposed causal reasoning framework itself. For instance, the method’s ability to handle ambiguous, non-deterministic, or cyclic reasoning paths is not evaluated. A more meaningful analysis should include failure cases that arise even when parsing is correct and the theorem exists, for example, when multiple conflicting causal paths are feasible or when causal deduction leads to incorrect intermediates due to compositional error propagation. Is the method robust to these intermediate error?
5. While the paper emphasizes interpretability and optimality, it lacks a complete worked example that would substantiate this claim. A concrete walk-through of a single example, showing the causal graph traversal, matrix deduction steps, and resulting multiple (or shortest) solutions, would strengthen the claims. For comparison, the cited work E-GPS provides detailed solution reasoning trees and theorem-level step tracing, which make interpretability more transparent. I recommend including a similar case in the main paper or appendix.

Subtle details:
1. The citation format in the paper looks strange and hard to follow. First, the references are not numbered in the appendix, making it hard to search for the corresponding reference. Second, isn’t the reference number should be wrapped in a square bracket []?
2. In section 3.2.1, line 151, the primitive node g is used before it got defined in the next paragraph.

---

> ### Author Rebuttal · Authors · 2025-07-31
>
> Q1: How does the proposed method perform on other popular datasets, such as FormalGeo7K and MATH? There should be exploration of robustness to noisy inputs, or evaluation on other datasets to test generalizability. Is the 24 hand-crafted symbolic rules brittle in other datasets?
>
> We understand that in most scenarios (e.g., involving neural networks), evaluation on more datasets would make the results more supportive (e.g., better generalizability). Here are the reasons why we only report results on Geometry3K in the first submission:
>
> — Geometry3K is a widely recognized and used data set that has provided both predicted and ground-truth formal language parsings. All problems in this dataset are well annotated and standard, covering a wide range of problem types, many of which are also selected as composition of other datasets (e.g., FormalGeo7K). Since previous baselines are all established and developed based on the same symbolic system (i.e., Inter-GPS style), it is a common sense for fair comparison on this dataset.
>
> — For symbolic-based methods, one of the intrinsic advantages is stableness. That means, if the input is sent in the pre-defined format, each step of symbolic deduction is promising and vigorous. Therefore, the generalizability mainly depends on geometry problem parsing, the design of symbolic system and theorem rule base. As the performances on Geometry3K shows, the theorem rule base has been developed as applicable to most of the geometry problem scenarios. This indicates its applicability to a wide range of standard geometry problems. Note that we do not augment the theorem rule base for fair comparison on solving performance.
>
> — FormalGeo7K is another widely used data set with formal language annotations, but in a different symbolic system design. It is now mainly used for neural-based methods and methods developed on its system. We have been making efforts during the rebuttal time to translate the system language, and sadly it is too much workload for us to implement a correct projection. Further, we have observed that FormalGeo7K test set includes a large portion of geometry problems selected from the Geometry3K train. Considering that they both are standard geometry problem type, we believe the results on Geometry3K has strong .
>
> — MATH dataset includes problems include but not limited to the topics of algebra, probability, calculus and a small portion of geometry. It is designed for LLM-integrated methods and evaluation of LLM’s math capabilities. It does not provide any parsing annotations in formal language. Therefore, it is not quite applicable for our task and its beyond our scope.
>
> We sincerely hope that you could reconsider the necessity of evaluation on these datasets for symbolic methods, taking into account the existing limitations of this task and public data.
>
> As for the robustness to noisy inputs, we have listed both results of ground-truth input and predicted input (i.e., with incorrect parsing results) in last two lines of Table 1. For all symbolic-based methods, the performance is sensitive to the correctness of input content. This has been analyzed and discussed in the third point of our quantitative analysis in section 4.2 (Line 314).
>
> As for the 24 hand-crafted symbolic rules, these theorem rules were meticulously designed, encompassing most geometry theorems required for common geometry problems in educational scenarios. A list of these theorems can be found in appendix C. Moreover, these rules have been validated to be effective for problem solving in previous works and our work (e.g., surpassing the problem solving accuracy obtained human experts). Therefore, for geometry problems in other datasets that are similar to those widely adopted in education, these rules are applicable.
>
> Q2: Ablation of Causal Graph Reasoning (CGR) and Forward Matrix Deduction (FMD).
>
> Thank you for pointing out this. We initially thought that it is not appropriate to compare the efficiency (e.g., time) of using and not using FMD, because the resource consumption is different. However, we find it still necessary to better support the introduction of FMD.
>
> First, we want to re-emphasize that Causal Graph Reasoning (CGR) is proposed as the theoretical foundation to achieve causal reasoning for solution exploration. FMD is one of possible detailed strategies that we propose to implement CGR. So it’s not simply an A+B module-combining architecture that supports independent ablations. We implemented a basic strategy, leveraging Python dictionaries for causal graph reasoning and node status storage. To provide a fair and detailed comparison, we record the average reasoning time per iteration of these two methods, grouped by the number of unique primitive nodes and the number of deduction paths (i.e., sum of all deduction paths in all causal graphs). The other parts and settings remain the same. Experiments were run for 3 times and the average result is reported.
>
> Grouped by number of all deduction paths (/s):
>
> | | [0, 100]  | (100, 1000]  | (1000, 2000]  | (2000, 5000]  | (5000, 10000] | (10000, )  |
> | :--- | :--- | :--- | :--- | :--- | :--- | :--- |
> | **Basic** | 1.57e-3 | 1.26e-2 | 9.62e-3 | 1.31e-2 | 2.60e-2 | 1.86e-2 |
> | **FMD** | 1.19e-3 | 1.70e-3 | 2.35e-3 | 6.46e-3 | 2.87e-2 | 7.76e-2 |
>
> Grouped by number of unique primitive nodes (/s):
>
> | | ≤20 | 20~30 | 30~50 | 50~70 | ≥70 |
> | :--- | :--- | :--- | :--- | :--- | :--- |
> | **Basic** | 1.23e-3 | 4.55e-3 | 1.02e-2 | 1.93e-2 | 4.41e-2 |
> | **FMD** | 1.19e-3 | 1.88e-3 | 4.11e-3 | 1.65e-2 | 6.86e-2 |
>
> From the results, we mainly have the following observations:
>
> (1) FMD consistently performs faster than using Basic when the number of deduction paths is smaller than 5,000 in table 1 and when the number of primitive nodes is smaller than 70 in table 2. Within this range, the time of FMD generally exhibits a steady increasing trend with the increasing of deduction paths and primitive nodes. When the sum of deduction paths increases to 10,000 and even greater, the dimensions of matrices will expand, resulting in more resource demands. Under such extreme and abnormal circumstances, which is not applicable to most geometry problems, it is recommended to simply use basic strategy of causal graph reasoning.
>
> (2) When the number of deduction paths exceeds 10,000, the time of basic strategy is surprisingly lower than when the number of paths ranges between 5,000 and 10,000. This is because in this kind of extreme samples, there are only few feasible deduction paths (i.e., successfully activate the path because head nodes and prerequisites are activated) even if there are many connected deduction paths.
>
> These experiments and according analysis will be added in final version.
>
> Q3: It is over-claimed that the method is general beyond geometry problem solving in the section 5, because no empirical or theoretical support outside the scope of GPS is discussed. This should be toned down.
> Thanks for your suggestion on this. Our motivation on this point is to inspire more researchers from domains where the task can be reformulated as such, especially for those relying on explicitly pre-defined action space and costly real action application. We have recognized that stating such conclusion might not be appropriate and we will revise the related content to present a more rational discussion in final version.
>
> Q4: More case studies with deeper analysis is needed for more comprehensive discussion of limitations of the method itself.
>
> Thanks for your precious advice. We will further add more cases and enrich related discussion in the final version from the potential dimension explosion aspect: in some extreme geometry problem sample, the diagram includes hundreds of unique primitives and thus largely increasing the composition of primitives and dimensions of matrices. In this situation, there is high risk for resource being exhaustive. We will try to find more cases to have deeper analysis.
>
> Q5: A complete worked example and a concrete walk-through this example could be included for better illustration.
>
> Thank you for this useful advice. We also believe that a walk-through example which is used to explain in details how the data is processed is beneficial for deeper understanding. In this submission, we simplified the matrix and depicted only partial matrix deduction details because the actual dimension of matrix is usually not appropriate for illustration. We will find a more suitable case for illustration of all the detailed procedures, e.g., step-by-step causal graph reasoning and forward matrix deduction. Please provide us this chance to present it in the final version. (Inclusion of figures in rebuttal is not allowed)
>
> Q6: The citation format in the paper looks strange and hard to follow.
>
> We are truly sorry for the inconvenience caused to your reading experience. We have also noted that the current reference format is not appropriate, as it uses the default plain style. It will be corrected and aligned with the official requirements in final version.
>
> Finally, we sincerely appreciate the invaluable suggestions you provided, as they really contribute to refining the presentation of our work. We also respectfully hope that you could kindly increase the score if most of your concerns are resolved.

---

> > ### Author Response · Authors · 2025-08-05
> > **We look forward to further discussions with you.**
> >
> > Dear reviewer, we appreciate your time and effort in reviewing our paper and providing valuable feedback. We are pleased to receive your positive assessment of our work's quality and novelty, and we are grateful for these constructive suggestions that will help improve our manuscript.
> >
> > If there's any specific point that needs further clarification and explanations, please kindly let us know. We are committed to addressing your remaining concerns and providing more details.
> >
> > Thank you.

---

> > > ### Comment · Reviewer_kUXM · 2025-08-06
> > >
> > > Thank you to the authors for responding thoughtfully to most of the concerns raised, especially around the overclaim of generality and the lack of ablation. I appreciate the clarification regarding dataset choices and the new ablation results comparing the causal graph reasoning and forward matrix deduction strategies. Your commitment to including a detailed walk-through example and improved formatting in the final version is encouraging.
> > >
> > > Some issues remain under-addressed, particularly the lack of robustness analysis and deeper exploration of the method’s limitations beyond input parsing. I hope the authors will include more comprehensive case studies and failure analyses in the camera-ready version.
> > >
> > > I have updated my overall assessment in light of the rebuttal and appreciate the improvements made.

---

> > > > ### Author Response · Authors · 2025-08-06
> > > >
> > > > We appreciate you taking the time in reviewing our paper and providing these valuable suggestions. Your recognition and support of our work have been a great encouragement to us. We will make sure to provide more comprehensive case studies and according analyses, as well as the detailed illustrations. Other modifications of the manuscript will also be updated appropriately in final version.
> > > >
> > > > Thank you.

---

### Official Review · Reviewer_8Ps7 · 2025-07-06

**Clarity:** 2
**Significance:** 3
**Originality:** 4
**Rating:** 5
**Confidence:** 3

**Summary:**

This paper proposes a graph-based framework for solving geometry problems, aiming to improve interpretability and generate shorter solutions. The method encodes the preconditions and reasoning rules of various theorems into a graph representation, and systematically applies iterative reasoning over this structure to derive solutions. To enhance efficiency, matrix operations are employed to accelerate the reasoning process. Empirical results demonstrate that the proposed approach achieves higher accuracy and produces shorter solutions compared to state-of-the-art methods.

**Questions:**

1. What is the runtime performance comparison between the proposed method and the baselines? Specifically, how much speedup is achieved by using matrix operations compared to the sequential reasoning process?
2. Could you clarify the evaluation metric: a problem is considered solved only if the answer obtained is the closest to the ground-truth? Does this imply that longer or substantially different, but still correct, solutions are not counted as successful?

**Ethical Concerns:**

["NO or VERY MINOR ethics concerns only"]

**Final Justification:**

The authors have adequately addressed the questions I raised, so I will keep my positive score unchanged.

**Limitations:**

Yes

**Paper Formatting Concerns:**

N.A.

**Quality:**

4

**Strengths And Weaknesses:**

#### Quality
The paper is of high quality. The proposed graph representation for solving geometry problems is both novel and well-grounded. Its transformation into matrix operations enables GPU acceleration, significantly improving reasoning speed. The methodology ensures interpretability of solutions, and the empirical evaluation demonstrates that the proposed approach produces shorter solutions.

#### Clarity

The paper is generally easy to follow, though certain aspects could be improved for better clarity.

I recommend bringing the overall framework into the main content to provide readers with a systematic understanding of the proposed method. Additionally, it should be clarified that the causal graph used here is not a simple graph, but a “hypergraph”, which supports edges from a set of vertices to another edge. A brief explanation distinguishing it from the causal graphs commonly used in causal inference would also help avoid confusion.

Some minor comments:
1. The citation format appears inconsistent. Papers are cited using indices, but the corresponding indices are missing in the reference list, and the brackets around the in-text citation numbers are also absent.
2. In Figure 1(c), please explain the red redundant steps, particularly the meaning of the numbers, e.g., 17, 1, 2, etc.
3. Consider briefly walking through Algorithm 1 to guide the reader.
4. In Equation 1, $E_{i,j}$ should be $R*$.
5. $g(k,l)$ is missing in Equation 3.
6. In Table 1, it would be helpful to specify that the numbers in the first few columns represent accuracy.

#### Significance
The proposed method is impactful. It not only produces shorter and interpretable solutions to geometry problems but also generates multiple solutions, which can be particularly valuable for educational purposes. Empirical results show that it outperforms both prior methods and human experts.

#### Originality
The idea of representing theorem rules as a graph and accelerating the reasoning process via matrix operations is novel and original to the best of my knowledge.

---

> ### Author Rebuttal · Authors · 2025-07-31
>
> We sincerely thank you for your support of our work. Your affirmation of our paper and novelty of the method provides much encouragement for us. Here are our responses to your concerns and questions.
>
> Q1: Suggestions on some professional term and position of overall framework.
>
> Yes, we agree that the overall framework is better to be placed in the methodology part in main content. In our initial submission, we utilized two separate, detailed figures to better illustrate our methods, consequently placing the framework in appendix due to page limit. Also, the term ‘hypergraph’ should be more exact than ‘graph’ and we will change it in our final version.
>
> Q2: Explanation of the red content in figure 1 (c).
>
> In case 1 of figure 1 (c), the numbers shown refer to the indices of theorem rules in the theorem rule base. The solution records the application sequence of theorem rules to get final answer, where the red numbers indicate redundant rules applied but yielded no benefit in answer acquisition. In case 2 of figure 1 (c), the tokens represent the operators and operations of an executable program sequence, where the red tokens represent redundant steps. For the problem in case 2, only one step is needed to get the final answer. Thanks for your question and we will add detailed explanations in the figure description in final version.
>
> Q3: Some mistakes in the equations and reference citation format.
>
> Thank you for pointing out our mistakes in the equations. We will thoroughly review the paper and ensure the correctness of equations, figures and other content in final version.
> We have also noted that the current reference format is not appropriate, as it uses the default plain style. It will be corrected and aligned with the official requirements in final version.
>
> Q4: The runtime performance compared with baselines and the ablation studies of the proposed Forward Matrix Deduction (FMD).
>
> We want to specify that a direct and thorough comparison of runtime with baselines may not be fair and applicable. On one hand, the goal and contribution is different, where our method aimed at exploration of explainable, shorter and multiple solutions. On the other hand, some baselines are not open-sourced and the resource consumption is varying. Therefore, we default to keep the time within acceptable range. As our method needs extra causal graph establishment and reasoning time, it does take longer time than the basic method Inter-GPS. Specifically, nearly half of problems can be solved within 1s, most problems can be solved within 5s, and only a small fraction of problems need more than 20s, where the method usually fails to obtain the answer.
>
> For more comprehensive analysis, we add the ablation studies of the proposed Forward Matrix Deduction (FMD). We want to re-emphasize that Causal Graph Reasoning (CGR) is proposed as the theoretical foundation to achieve causal reasoning for solution exploration. FMD is one of possible detailed strategies that we propose to implement CGR. So it’s not simply an A+B module-combining architecture that supports independent ablations. We implemented a basic strategy, leveraging Python dictionaries for causal graph reasoning and node status storage. To provide a fair and detailed comparison, we record the average reasoning time per iteration of these two methods, grouped by the number of unique primitive nodes and the number of deduction paths (i.e., sum of all deduction paths in all causal graphs). The other parts and settings remain the same. Experiments were run for 3 times and the average result is reported.
>
> Grouped by number of all deduction paths (/s):
>
> | | [0, 100]  | (100, 1000]  | (1000, 2000]  | (2000, 5000]  | (5000, 10000] | (10000, )  |
> | :--- | :--- | :--- | :--- | :--- | :--- | :--- |
> | **Basic** | 1.57e-3 | 1.26e-2 | 9.62e-3 | 1.31e-2 | 2.60e-2 | 1.86e-2 |
> | **FMD** | 1.19e-3 | 1.70e-3 | 2.35e-3 | 6.46e-3 | 2.87e-2 | 7.76e-2 |
>
> Grouped by number of unique primitive nodes (/s):
>
> | | ≤20 | 20~30 | 30~50 | 50~70 | ≥70 |
> | :--- | :--- | :--- | :--- | :--- | :--- |
> | **Basic** | 1.23e-3 | 4.55e-3 | 1.02e-2 | 1.93e-2 | 4.41e-2 |
> | **FMD** | 1.19e-3 | 1.88e-3 | 4.11e-3 | 1.65e-2 | 6.86e-2 |
>
> From the results, we mainly have the following observations:
>
> (1) FMD consistently performs faster than using Basic when the number of deduction paths is smaller than 5,000 in table 1 and when the number of primitive nodes is smaller than 70 in table 2. Within this range, the time of FMD generally exhibits a steady increasing trend with the increasing of deduction paths and primitive nodes. When the sum of deduction paths increases to 10,000 and even greater, the dimensions of matrices will expand, resulting in more resource demands. Under such extreme and abnormal circumstances, which is not applicable to most geometry problems, it is recommended to simply use basic strategy of causal graph reasoning.
>
> (2) When the number of deduction paths exceeds 10,000, the time of basic strategy is surprisingly lower than when the number of paths ranges between 5,000 and 10,000. This is because in this kind of extreme samples, there are only few feasible deduction paths (i.e., successfully activate the path because head nodes and prerequisites are activated) even if there are many connected deduction paths.
> These experiments and according analysis will be added in final version.
>
> Q5: Could you clarify the evaluation metric: a problem is considered solved only if the answer obtained is the closest to the ground-truth? Does this imply that longer or substantially different, but still correct, solutions are not counted as successful?
>
> No. Under this metric setting, all solutions of all methods that finally lead to the correct answers are counted as correct, regardless of the lengths. We consider these solutions correct in order to calculate the final solving performance.
>
> Finally, we again appreciate your support for our work and it gives us much encouragement.

---

> > ### Comment · Reviewer_8Ps7 · 2025-08-05
> > **Questioning the Effectiveness of FMD**
> >
> > Thank you for your clarification and for providing the additional experimental results.
> >
> > However, I’m still unclear about the purpose of the Forward Matrix Deduction (FMD) strategy. My understanding was that FMD was introduced to accelerate the causal graph deduction process, especially in scenarios where the runtime might otherwise become prohibitively high.
> >
> > Yet, the two tables you provided indicate that FMD’s runtime is generally comparable to that of the Basic strategy, and in all cases, both strategies complete within a reasonable time. More importantly, as the problem size increases (i.e., with more deduction paths and a higher number of unique primitive nodes), the FMD strategy appears to perform even slower than the Basic strategy.
> >
> > Given these observations, the contribution and practical benefit of FMD remain unclear to me. Could you please clarify its intended advantage or explain in which scenarios FMD demonstrates clear superiority?

---

> > > ### Author Response · Authors · 2025-08-06
> > >
> > > Thank you for responding to us. Here we provide a more detailed justification for the introduction of FMD, considering the methodological motivations and its significance in real-world applications.
> > >
> > > From the methodological perspective, FMD leverages the matrix manipulation and consolidates the sequential condition check into one single operation. This means, regardless of the number of head nodes and prerequisites in a causal deduction path, FMD is able to produce the deducted results at a similar speed. The percentage of involved nodes dose not affect the speed of each matrix deduction, because all nodes are already involved. As for the basic strategy, this traversal status checking manner is bound to become increasingly time-consuming as the number of involved head nodes and prerequisites grows. For example, if a deduction path requiring $n$ head nodes and prerequisites takes $x$ seconds when using basic strategy, it may take up to $2x$ seconds if another deduction path requires $2n$ head nodes and prerequisites. In contrast, FMD handles the two deduction paths with the same computational efficiency. Additionally, from the tables we provided in rebuttal, there’s actually clear difference between FMD and Basic when the number of primitive nodes falls between 0~70 and number of all deduction paths range from 0 to 5,000. The speed gap will be further widened when dealing with more complex scenarios where the deduction paths involve higher percentage of nodes. For extreme cases with more than 10,000 deduction paths, which are uncommon in real life, most of the paths are actually invalid (i.e., unable to activate). According to our implementations, these paths are simply skipped in basic strategy, while they are not eliminated in FMD, occupying the space and reducing the speed. It can be further modified by matrix manipulation refinement techniques (we did not adopt any optimization techniques to matrix manipulations in this version). Therefore, FMD holds great potential for future development in complex scenarios (i.e., scenarios with a higher degree of node involvement), and provides certain insights for similar scenarios in other fields.
> > >
> > > From the practical application perspective, FMD also presents inherent advantages over the basic strategy. On one hand, the comparison tables in our rebuttal record the per-iteration results of two strategies. The accumulated time difference will become larger when the number of iterations increases. Particularly, the actual iterations required is usually unknown and could be large in real-world applications and in other scenarios. On the other hand, FMD enables us to transfer the reasoning process from cpu to gpu, sharing the computational load when cpu is heavily utilized. Note that the comparison results are obtained when processing the problems sequentially. When it comes to practical scenarios, a significant number of users may post geometry problems for solving. If all procedures are processed by cpu, the speed will inevitably be slower and cost more time, compared to single-problem cases. Under such circumstances, distributing the computational workload of cpu by staggering the reasoning processes is meaningful.
> > >
> > > We hope these clarifications could resolve your concerns. Thank you.

---

> > > > ### Comment · Reviewer_8Ps7 · 2025-08-07
> > > >
> > > > Thank you for your clarification. I suggest including the justification for FMD in your paper, along with the empirical comparison to support your claims. Additionally, please consider providing experimental evidence that demonstrates the superior of FMD over the Basic strategy in the following specific scenario you mentioned.
> > > >
> > > > “As for the basic strategy, this traversal status checking manner is bound to become increasingly time-consuming as the number of involved head nodes and prerequisites grows.”
> > > >
> > > > I will keep my score unchanged for now.

---

> > > > > ### Author Response · Authors · 2025-08-08
> > > > >
> > > > > We appreciate you taking the time in reviewing our paper and providing these valuable suggestions. We will enrich the discussion about the introduction of FMD, as well as these experimental evidences. Other modifications of the manuscript will also be updated appropriately in final version.
> > > > >
> > > > > Thank you.

---

### Note · Authors · 2025-08-12

Dear Area Chairs and all five reviewers,

We sincerely appreciate your time and effort in reviewing our paper. We are greatly encouraged by the reviewers' recognition of the quality and novelty of our work, and are gratified by the acknowledgement of the potential impact of our work in the geometry problem solving domain and even beyond. More importantly, your valuable suggestions provided during the review and rebuttal process have been instrumental in further refining our manuscript and enhancing its comprehensiveness. After the discussion phase, we are delighted to see that most concerns have been solved, leading to a consensus from all reviewers for the acceptance of our work.

We will ensure the changes discussed in rebuttal phase to be appropriately incorporated and revised in final version of our manuscript.

Thank you.

---

### Decision · Program_Chairs · 2025-09-17

**Decision:**

Accept (poster)

**Comment:**

**Claims and findings of the paper**

This paper introduces Causal-R, a novel symbolic reasoning framework for solving geometry problems. The core contributions are the "Causal Graph Reasoning" (CGR) theory, which models theorem-based deductions as causal paths, and an efficient implementation called "Forward Matrix Deduction" (FMD) that uses matrix operations to explore the solution space. The authors claim this approach effectively compresses the search space, enabling the generation of multiple, shorter, and more interpretable solutions. On the standard Geometry3K benchmark, the proposed method is shown to outperform existing symbolic and neural baselines in both accuracy and solution length.

**Strengths of the paper**

*   The paper's central idea of reformulating geometric reasoning as a causal deduction process on a graph is novel and well-motivated. This approach provides a new perspective on symbolic problem-solving that moves beyond simple forward/backward chaining. The reviewers unanimously praised the originality of the framework (8Ps7, kUXM, SAZz, 57ZD).
*  The method achieves state-of-the-art results on the Geometry3K dataset, demonstrating higher accuracy and producing significantly shorter solutions than prior methods and even human experts. This provides strong evidence for the effectiveness of the proposed approach (8Ps7, kUXM, VP5W).
*   By design, the solutions generated by Causal-R are interpretable, as each step corresponds to a specific geometric theorem. The ability to generate multiple, shorter solutions is a significant advantage, particularly for potential applications in education (8Ps7).
*  The paper is generally well-written and clearly presents its motivation, methodology, and results, making it accessible even to readers not deeply familiar with the subfield (8Ps7, SAZz, 57ZD).

**Weaknesses of the paper**

*  A primary concern raised by reviewers (kUXM, VP5W) is that the evaluation is confined to a single dataset (Geometry3K). The paper lacks experiments on other benchmarks (e.g., FormalGeo7k) or a robustness analysis, which limits claims of generalizability.
*   While FMD is presented as a key contribution for efficiency, the ablation study provided during the rebuttal showed that it can be slower than a basic implementation on problems with a very large number of deduction paths. The authors' justification focused on methodological potential (e.g., GPU acceleration, handling complex prerequisites) rather than demonstrating a consistent, practical speedup in the current experiments (8Ps7).
*   The use of the term "causal" was questioned (SAZz) as it does not align with the formalisms of mainstream causal inference literature (e.g., Pearlian frameworks). While the authors clarified their definition is specific to deductive necessity within theorems, this could be a source of confusion.
*   The initial submission lacked crucial ablation studies and contained some overstated claims regarding generality and solution "minimality." These were largely addressed during the rebuttal but were significant initial weaknesses.

**Key reasons to accept**

The paper presents a solid, novel, and impactful contribution to the field of automated mathematical reasoning. The proposed Causal-R framework is elegant and demonstrates impressive performance on a standard benchmark. The authors' diligent and effective engagement during the rebuttal period has addressed the most critical weaknesses of the initial submission and gives confidence that the final version will be of high quality. Despite the limited scope of the evaluation, the core ideas are strong and likely to inspire future work in symbolic reasoning. Therefore, the paper is a clear accept.